

# Understanding monsoon controls on the energy and mass balance of Himalayan glaciers

*Authors:* Stefan Fugger[1,2], Catriona L. Fyffe[1,3], Simone Fatichi[4], Evan Miles[1], Michael McCarthy[1,5], Thomas E. Shaw[1], Baohong Ding[6], Wei Yang[6], Patrick Wagnon[7], Walter Immerzeel[8], Qiao Liu[9], and Francesca Pellicciotti[1,3]

[1]Swiss Federal Institute for Forest, Snow and Landscape Research (WSL),Birmensdorf, Switzerland
[2]Institute of Environmental Engineering, ETH Zurich, 8093 Zurich, Switzerland
[3]Department of Geography and Environmental Sciences, Northumbria University, Newcastle, UK
[4]Department of Civil and Environmental Engineering, National University of Singapore, Singapore
[5]British Antarctic Survey, Cambridge, UK
[6]Institute of Tibetan Plateau Research, Chinese Academy of Sciences, Beijing, China
[7]Univ. Grenoble Alpes, CNRS, IRD, Grenoble-INP, IGE, 38000 Grenoble, France
[8]Faculty of Geosciences, Department of Physical Geography, Utrecht University, Utrecht, The Netherlands
[9]Institute of Mountain Hazards and Environment, Chinese Academy of Sciences, Chengdu, China

**Correspondence:** Stefan Fugger (stefan.fugger@wsl.ch)

**Abstract.** The Indian and East Asian Summer Monsoons shape the melt and accumulation patterns of glaciers in High Mountain Asia in complex ways due to the interaction of persistent cloud cover, large temperature amplitudes, high atmospheric water content and high precipitation rates. While the monsoons dominate the climate of the southern and eastern regions, they progressively lose strength westward towards the Karakoram, where the influence of Westerlies is predominant. Despite the

major role of the monsoon in the Himalayas, a holistic understanding of their influence on the region's glaciers is lacking because previous applications of energy- and mass-balance models have been limited to single study sites. In this study, we use a full energy- and mass-balance model and seven on-glacier automatic weather station datasets from different parts of the Himalayas to investigate how monsoon conditions influence the glacier surface energy and mass balance. In particular, we look at how debris-covered and debris-free glaciers respond differently to monsoonal conditions. The radiation budget mostly controls

the melt of clean-ice glaciers, but turbulent fluxes also play an important role in modulating the melt energy on debris-covered glaciers. The sensible heat flux reduces during core monsoon, but the latent heat flux removes energy from the surface due to evaporation of liquid water. This interplay of radiative and turbulent fluxes, together with compensations between increasing and decreasing melt rates over the diurnal cycle, causes debris-covered glacier melt rates to stay almost constant over the entire ablation period through the different phases of the monsoon. Ice melt under thin debris, on the other hand, is amplified by both

the dark surface albedo and the turbulent fluxes, which act as a source of energy through surface heating and condensation, especially during monsoon. Pre-monsoon snow cover can considerably delay melt onset and have a strong impact on the seasonal mass balance. Intermittent monsoon snow cover further modulates the melt rates at high elevation. Given our results, we expect the mass balance of debris-covered glaciers to react less sensitively to projected future monsoon conditions than clean-ice and





dirty-ice glaciers. This work is fundamental to the understanding of the present and future Himalayan cryosphere and water
budget evolution, while informing and motivating further glacier- and catchment-scale research using process-based models.
    (Yang et al., 2017)

# 1  Introduction

High Mountain Asia (HMA) holds the largest ice volume outside the polar regions (Farinotti et al., 2019) and due to the large
elevation range and vast geographic extent, HMA glaciers are highly diverse in character and hydro-climatic situation (Yao
et al., 2012). Several large-scale weather patterns interact with the region's topography (Bookhagen and Burbank, 2010), caus-
ing glaciers to contrast in terms of hypsometry (Scherler et al., 2011a) and accumulation and ablation seasonality (Maussion
et al., 2014). The Indian Summer Monsoon dominates the Central Himalaya and the Southeastern Tibetan Plateau during sum-
mer, and gradually loses strength moving towards the Karakoram, Pamir and Kunlun ranges in the east, where the influence
of Westerlies is particularly strong. A more continental, monsoon-westerlies-influenced regime controls the Central Tibetan
Plateau (Yao et al., 2012; Mölg et al., 2014), and the East Asia Monsoon influences the eastern slopes of the Tibetan Plateau
(Maussion et al., 2014; Yao et al., 2012). These major modes of atmospheric circulation do not only control the surface pro-
cesses and runoff regimes of glaciers (e.g. Mölg et al., 2014, 2012; Kaser et al., 2010) but also lead to distinct responses of
glaciers to climate change (Yao et al., 2012; Sakai and Fujita, 2017; Scherler et al., 2011b; Kraaijenbrink et al., 2017). For ex-
ample, mass losses are high throughout most of HMA, and are particularly pronounced on the South-Eastern Tibetan Plateau,
while glaciers exhibit a near-balance regime in the Karakoram, Pamir and Kun Lun (Farinotti et al., 2020; Gardelle et al., 2012;
Brun et al., 2017; Shean et al., 2020).
    Accurate glacier mass balance modelling is essential to assess glacier meltwater contribution to mountain water resources,
and to predict future glacier states and catchment runoff. Physically-based models of glacier energy- and mass balance repre-
sent surface and sub-surface energy fluxes using physical equations to calculate the energy residual, i.e. the energy available for
melt, and the glacier runoff. They have provided an understanding about the individual processes controlling the glacier mass
balance and the climatic sensitivity of glaciers in their specific hydro-climatic environment (e.g. Mölg et al., 2012). Summer-
accumulation type glaciers in HMA experience simultaneous accumulation and ablation and their mass balances are known to
be highly sensitive to climatic variability during the monsoon season (Fujita and Ageta, 2000), when warm air temperatures
and high moisture influx coincide. Using energy balance modelling for an inter-annual study for the Central Tibetan Zhadang
glacier, Mölg et al. (2012) demonstrated that the timing of monsoon onset and the associated albedo variability can change
melt-rates substantially in subsequent years. At the same time, they observed a de-coupling of the glacier mass balance from
the Indian Summer Monsoon's control during the core monsoon season. Mölg et al. (2014) explain the mass balance variability
of Zhadang Glacier as being controlled by both the Indian Summer Monsoon onset and remotely controlled by mid-latitude
Westerlies. Combining energy balance with weather forecast modelling, Bonekamp et al. (2019) identify the timing and quan-
tity of snowfalls as the main source of differences in mass-balance regimes between the Shimshal Valley in the Karakoram and



the Langtang Valley in the Central Himalaya. Similarly, Zhu et al. (2018) attribute mass balance differences of three glaciers on the Tibetan Plateau mainly to different local rain-/snowfall ratios and timing.

The presence of debris cover, a widespread characteristic of HMA glaciers, (e.g. Herreid and Pellicciotti, 2020; Kraaijenbrink et al., 2017; Scherler et al., 2011b), creates additional complexity to understanding and modelling the processes leading
to (sub-debris) glacier melt. In recent years, much effort has gone into developing energy balance models for debris-covered glaciers, (e.g. Fujita et al., 2014; Reid and Brock, 2010; Nicholson and Benn, 2006; Lejeune et al., 2013; Rounce et al., 2015; Evatt et al., 2015; Collier et al., 2014; Steiner et al., 2018). Yang et al. (2017) compares the energy balance of a debris-covered and a clean-ice glacier on the Southeastern Tibetan Plateau and finds the main differences, beside the differences in melt rates, is their climatic sensitivity and the important role of turbulent fluxes on debris-covered glaciers. Thick debris is a stronger
control on melt rates than elevation (Shah et al., 2019) and also dampens and delays glacier melt in the diurnal cycle (Shrestha et al., 2020), while melt enhancement can occur where there is very thin or patchy debris (Fyffe et al., 2020). Ablation is often expected to be higher on such 'dirty ice glaciers' than at both clean-ice sites and at sites with established debris cover, as shown experimentally (Reznichenko et al., 2010; Östrem, 1959), and by means of modelling (Reid and Brock, 2010), with humidity being a determining factor for this enhancement (Evatt et al., 2015). Moisture in debris is an important factor under monsoonal
conditions, controlling the debris' thermal properties and thus ablation (Sakai et al., 2004; Nicholson and Benn, 2006) and has been the focus of devoted modelling studies (Giese et al., 2020; Collier et al., 2014). Moreover, the representation of latent heat due to evaporation (Giese et al., 2020; Steiner et al., 2018) and atmospheric stability correction for turbulent fluxes were shown to be important to improve the simulation of sub-debris melt (Reid and Brock, 2010; Mölg et al., 2012). Model implementation, however, remains complex and difficult to validate and transfer. Nevertheless, Nicholson and Stiperski (2020) showed that the
turbulent conditions over debris-covered and clean-ice sites are similar enough to be numerically treated in similar ways, i.e. the Monin-Obukhov similarity theory can be leveraged in the same way for the calculation of the turbulent fluxes, but with different parameters.

Observations of glacier surface meteorology, a prerequisite for accurate energy balance modelling, exist only for a few glaciers in HMA, and even fewer records exist for debris-covered glaciers. Direct observations of glacier mass balance in HMA also
remain sparse, and remote sensing observations are hindered by heavy cloud cover during the monsoon season.

Previous studies explicitly dealing with the imprint of the monsoon on the surface thermal properties of glaciers remained limited to individual clean-ice glaciers in the Central Tibetan Plateau (Mölg et al., 2012, 2014). Our main goal is to improve understanding of monsoon controls on glaciers in the Himalayan region by leveraging available automatic weather station (AWS) records in the region. Our specific objectives are: 1) to understand, in detail, the glacier energy and mass balance and
related controls at seven Himalayan study sites in a robust and systematic manner; 2) to assess the effects of monsoonal conditions on debris-covered and clean ice glaciers; 3) to investigate whether these effects are generalisable; and 4) to discuss possible implications of our findings under climate change. We address these objectives by applying the glacier energy and mass balance module of a land surface model suited to both debris-covered and clean-ice glaciers. We apply the model at the point scale of individual AWSs, driven by high-quality in situ meteorological observations that guarantee accurate energy
balance simulations, not affected by extrapolation of the meteorological forcing. By identifying and discussing the key surface





processes of glaciers and their dynamics under monsoonal conditions, this study promotes their appropriate representation in models of glacier mass balance and the hydrology of glacierised catchments.

## 2   Study sites and data

### 2.1   Sites and observations

In situ observations from seven on-glacier automatic weather stations in different environments along the climatic gradient of the Himalayas were gathered and are used for forcing and evaluation of the model (Table 2). Our seven study sites are located in the Central and Eastern Himalayas and cover a range of glacier types and local climates (Figure 1, 2 and Table 1). The seven sites include both spring- (24K, Parlung No.4) and summer-accumulation glaciers (all others) as indicated by the proportion

of monsoon precipitation to the annual precipitation (Figure 3). Langtang, Lirung and Yala are neighboring glaciers found in the Langtang Valley (Figure 1), which has an extensive history of glaciological and hydrological reasearch. The Langtang Valley is strongly influenced by the Indian Summer Monsoon (∼ June to October), during which more than 70% of the annual precipitation arrives (Figure 3 and Table 1), while the period from November to May is a drier season (Collier and Immerzeel, 2015; Immerzeel et al., 2012). The Langtang Valley has been a site of extensive glaciological (e.g. Fujita et al., 1998; Stumm

et al., 2020), meteorological (Immerzeel et al., 2014; Heynen et al., 2016; Steiner and Pellicciotti, 2016; Bonekamp et al., 2019; Collier and Immerzeel, 2015) and hydrological (e.g. Ragettli et al., 2015) investigations. On-glacier AWSs were installed during the ablation season on Lirung (2012-2015) and Langtang (2019) glaciers, and year-round on Yala (2012-ongoing) (Table 2). Both Lirung and Langtang are valley glaciers that have heavily debris-covered tongues, but the tongue of Lirung has disconnected from the accumulation zone (Figure 2). Yala is a considerably smaller clean-ice glacier, with most of its ice mass

located at comparably high elevation. It is oriented to the southwest and has a gentle slope (Fujita et al., 1998) (Figure 2 and Table 1).

North Changri Nup Glacier (hereafter Changri Nup Glacier) is a debris-covered valley glacier located in the Everest region in Nepal (Figure 1). The southeast-oriented, avalanche-fed glacier discharges into the Koshi River system. The local climate is similar to the one of the Langtang Valley, with 70-80% of precipitation falling during monsoon (Vincent et al., 2016) (Figure

2, 3 and Table 1).

24K and Parlung No.4 glaciers are located on the southeastern Tibetan Plateau, feeding water into the upper Parlung Tsangpo, a major tributary to the Yarlung Tsangpo - Brahmaputra River. The summer climate is characterized by monsoonal air masses reaching the Gangrigabu mountain range from the south through the Yarlung Tsangpo Grand Canyon. 24K Glacier is an avalanche fed valley glacier with a debris-covered tongue, located 24 km from the town of Bome (Yang et al., 2017). It is small

in size, oriented to the northwest and surrounded by shrubland (Figure 1, 2 and Table 1). Parlung No.4 is a debris-free valley glacier, which is north-east oriented, considerably larger than 24K and located 130km (south-east) from Bome (Yang et al., 2011) (Figure 1 and Table 1). Full automatic weather stations were installed in the ablation zones of both glaciers in 2016 and subsequent years (Table 2).

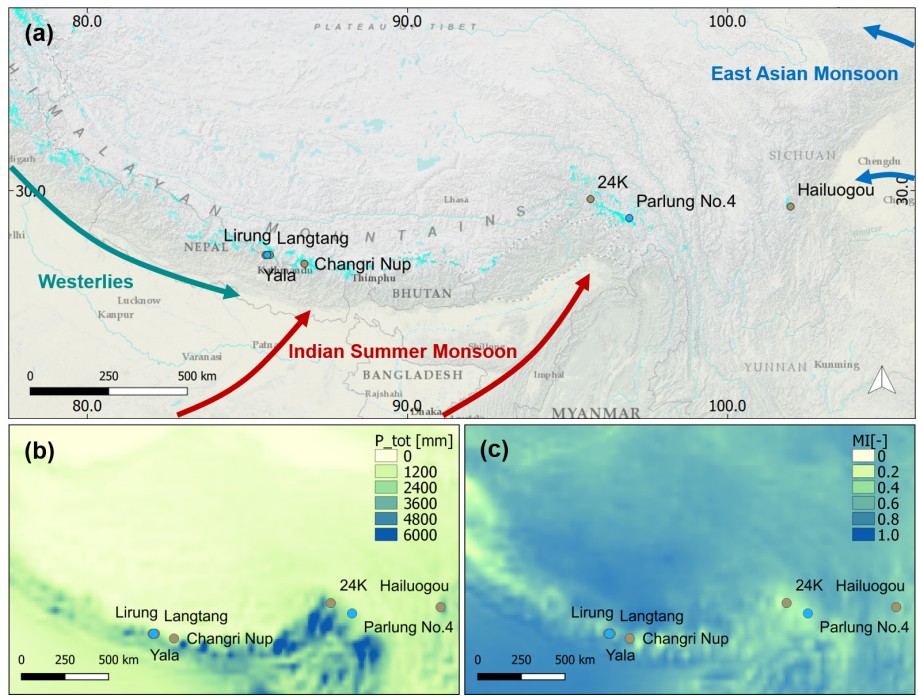

**Figure 1. (a)** shows the context of study sites with respect to large-scale weather patterns, topography and glacier distribution (turqoise, source: Randolph Glacier Inventory 6.0). Blue dots indicate clean-ice study glaciers and brown dots indicate debris-covered study glaciers. **(b)** displays the spatial pattern of average annual precipitation from ERA5-Land (1981-2019). **(c)** shows the monsoonal (June-September) portion of the ERA5-Land total annual precipitation (MI). Background map source: ESRI, U.S. National Park Service.

Hailuogou Glacier, the second-largest of our study sites (Figure 2) is located on the eastern slope of Mt. Gongga in the east-
ernmost portion of the southeastern Tibetan Plateau (Figure 1). It is located at low elevation and covered with thin and patchy debris, leading to high annual ablation rates (Figure 2 and Table 1). The local climate influenced by the East Asia Monsoon with typically only 50 to 60% of the annual precipitation arriving during the monsoon period (Figure 1 and 3). The debris-covered tongue is connected to a steep and extensive accumulation zone via a large icefall, but avalanching is the primary mass supply mechanism through the icefall to the valley tongue (Liao et al., 2020), and a dynamic disconnect is expected to occur in
the near future. Weather stations were installed at three nearby off-glacier locations and one on-glacier site during 2008, while precipitation was measured at Alpine Ecosystem Observation and Experiment Station of Mt. Gongga, within $1.5\,km$ from the glacier terminus (Table 2).

## 2.2 Climatic and meteorological conditions

Here, we use the monthly averaged ERA5-Land reanalysis data (Muñoz Sabater, 2019) to provide an overview of the long term climatic patterns, e.g. the average monsoonal regime from June through September, and evaluate the representativeness



**Table 1.** Characteristics of the study sites. Planimetric glacier and debris surface areas, mean elevation, slope and aspect were calculated using the updated Randolph Glacier Inventory 6.0 by Herreid and Pellicciotti (2020) and the USGS GTOPO30 digital elevation model. Slope and aspect are mean values for the whole glacier. MI ('Monsoon-Index') is the mean June-September portion of the ERA5-Land total annual precipitation (1981-2019)

| | Area [$km^2$] | | Elevation [$m.asl$] | | | Slope | Aspect | MI |
|---|---|---|---|---|---|---|---|---|
| | Glacier | Debris | min | max | mean | [°] | [°] | [-] |
| **Lirung** (LIR) | 4.0 | 1.5 | 3990 | 6830 | 5100 | 27.6 | 153.1 | 0.74 |
| **Langtang** (LAN) | 37.0 | 17.8 | 4500 | 6620 | 5330 | 16.0 | 175.2 | 0.71 |
| **Yala** (YAL) | 1.4 | - | 5170 | 5660 | 5390 | 23.5 | 228.2 | 0.74 |
| **Changri Nup** (CNU) | 2.7 | 1.4 | 5270 | 6810 | 5610 | 15.9 | 183.2 | 0.76 |
| **24K** (24K) | 2.0 | 0.9 | 3910 | 5070 | 4350 | 18.3 | 273.4 | 0.46 |
| **Parlung No.4** (NO4) | 11.0 | - | 4620 | 5950 | 5330 | 17.1 | 152.6 | 0.40 |
| **Hailuogou** (HAI) | 24.5 | 4.1 | 2980 | 7470 | 5360 | 27.0 | 117.5 | 0.56 |

**Table 2.** Summary of available meteorological and ablation observations at each site, as well as each site's model period. Variables indicated with * were transferred from neighboring weather station. Variables with ** were reconstructed based on other variables measured at the same station.

| | AWS Location | | | | Variables measured | | | Model period | Reference |
|---|---|---|---|---|---|---|---|---|---|
| | Lat | Lon | Elevation [m.a.s.l.] | Debris thickness [cm] | AWS | Precipitation | Ablation | begin/ end | |
| **Lirung** | 28.233 | 85.562 | 4076 | 30 | $T, RH, W_s, W_d, SW_\downarrow,$ $SW_\uparrow, LW_\uparrow, LW_\downarrow, P_{atm}^*$ | Pluvio Kyanging and Yala Basecamp, 3857m.asl and 5090m.asl, hourly, partly lapsed | SR50 | 2014-05-05/ 2014-10-24 | Ragettli et al. (2015) |
| **Langtang** | 28.279 | 85.722 | 4557 | 50 | $T, RH, W_s, W_d, SW_\downarrow,$ $SW_\uparrow, LW_\downarrow^{**}, LW_\uparrow, P_{atm}$ | Pluvio Morimoto base camp 4919m.asl, hourly | SR50 | 2019-05-11/ 2019-10-30 | unpublished |
| **Yala** | 28.233 | 85.612 | 5090 | - | $T, RH, W_s, W_d, SW_\downarrow,$ $SW_\uparrow, LW_\uparrow, LW_\downarrow, P_{atm}^*$ | Pluvio Yala base camp 5090m.asl, hourly | SR50 | 2019-05-01/ 2019-10-31 | ICIMOD (2021) |
| **Changri Nup** | 27.993 | 86.780 | 5471 | 10 | $T, RH, W_s, W_d, SW_\downarrow,$ $SW_\uparrow, LW_\uparrow, LW_\downarrow, P_{atm}^*$ | Pluvio at Pyramid meteorological station (4993 m a.s.l.), 4.9 km SE of AWS location, hourly | SR50 | 2016-05-01/ 2016-10-31/ | Wagnon (2021) |
| **24K** | 27.983 | 86.778 | 5362 | 20 | $T, RH, W_s, W_d, SW_\downarrow,$ $SW_\uparrow, LW_\uparrow, LW_\downarrow, P_{atm}^*$ | On-glacier tipping bucket, hourly | stake | 2016-06-01/ 2016-09-29 | Yang et al. (2017) |
| **Parlung No.4** | 29.761 | 95.720 | 3900 | - | $T, RH, W_s, W_d, SW_\downarrow,$ $SW_\uparrow, LW_\uparrow, LW_\downarrow, P_{atm}^*$ | Pluvio, near terminus 4600m.asl, hourly | stake | 2016-05-01 2016-10-31 | Yang et al. (2017) |
| **Hailuogou** | 29.558 | 101.969 | 3550 | 1 | $T, RH, W_s, W_d, SW_\downarrow,$ $SW_\uparrow, LW_\uparrow, LW_\downarrow, P_{atm}^*$ | Pluvio at GAEORS station, 3000m.asl, 1.5km from terminus | stake | 2008-05-15 2008-10-31 | Zhang et al. (2011) |

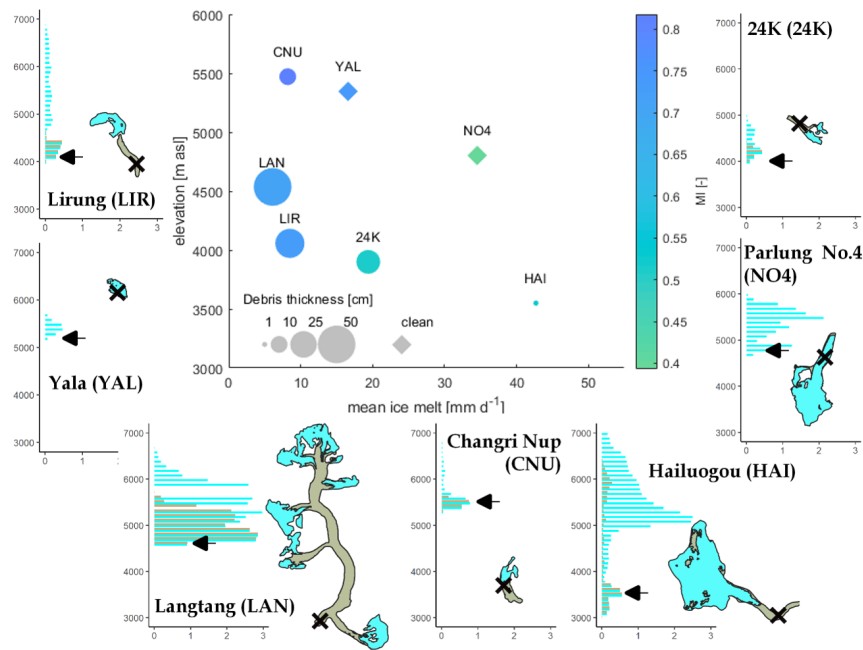

**Figure 2.** Characteristics of study sites, summarized (center) in terms of elevation, mean measured ice melt rate, measured debris thickness and JJAS contribution to the ERA5-Land total annual (1981-2019) precipitation (monsoon index; MI). For each site, we also show glacier (blue bars) and debris (brown bars) hypsometry, with area on the x-axis [$km^2$] and altitude on the y-axis [$m.asl$], and glacier and supraglacial debris extents. Black crosses and arrows indicate AWS location on the glacier.

of the AWS records in terms of seasonal variability (Figures A1 to A7), while acknowledging that the absolute values from the reanalysis dataset might be biased.

Incoming shortwave radiation (Figure 3b) shows a clear peak before monsoon onset at all sites. A smaller secondary peak is
reached just after the monsoon in October at the Central Himalayan sites, but not at the Eastern Himalayan sites. Interruptions in monsoonal overcast conditions (break periods) seem to be more common at the eastern sites, leading to occasional secondary peaks in incoming shortwave radiation during monsoon.

Average mean monthly 2 m air temperatures have a similar pattern at all study sites (Figure 3a), with a slow increase from January to a peak between July and August, just after peak monsoon, and a steeper decline from post-monsoon into winter. A
similar regime is followed by $LW_\downarrow$, with highest values reached during the core monsoon (Figure 3c).

There is a clear difference in the seasonal evolution of precipitation between the Central (Lirung, Lantang, Yala, Changri Nup) and the Eastern Himalayan sites (24K, Parlung No.4, Hailuogou) (Figure 3e): relatively high mean monthly precipitation during the monsoon period is contrasted by comparably low precipitation outside of this period. The eastern sites have less pronounced monsoonal precipitation peaks, and more gradual changes in precipitation intensities over the annual cycle. The Parlung sites
(24K and Parlung No.4) have two precipitation peaks: during spring and monsoon. Hailuogou exhibits the smoothest evolution

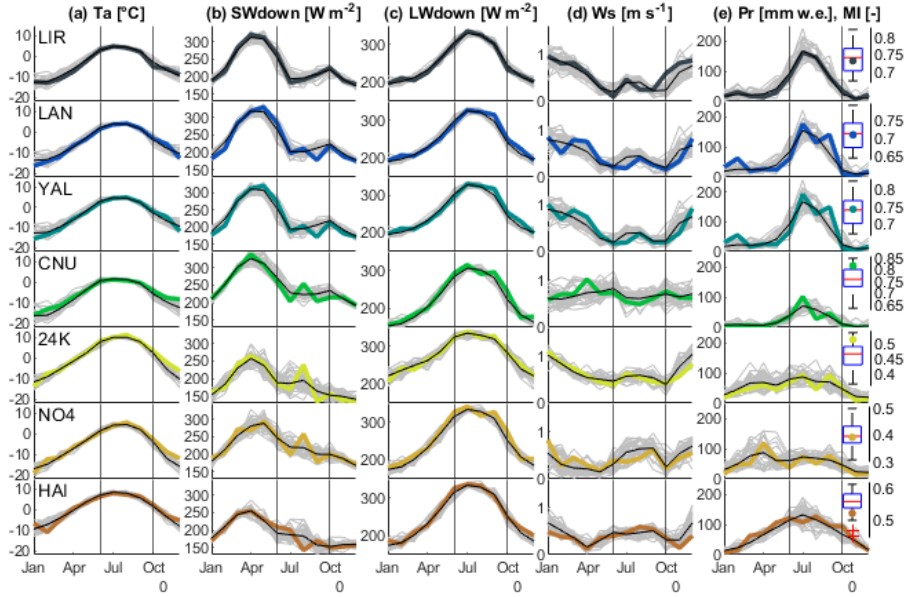

**Figure 3.** Monthly climatology derived from ERA5-Land for 1981-2019 (grey background lines), along with the monthly averages (black lines) and the study year at each glacier (colored lines). Plotted meteorological variables are **(a)** mean air temperature ($Ta$), **(b)** incoming shortwave radiation ($SW_\downarrow$), **(c)** incoming longwave radiation ($LW_\downarrow$), **(d)** wind speed ($Ws$) and **(e)** monthly precipitation sums ($Pr$). Black vertical lines indicate the average region-wide monsoon season. Boxplots show the monsoon index ($MI$) over ERA5-Land period as the fraction of monsoonal (June-September) to annual precipitation, with the colored dot indicating the value for the study year.

over the annual cycle with a clear maximum in July. A simple monsoon index (MI) is calculated for each year including the study year as the ratio between monsoon precipitation and annual average precipitation (Figure 3e). This value tends to be higher in the Central Himalaya compared to the sites on the South-Eastern Tibetan Plateau.

The yearly cycle of wind speeds (Figure 3d) varies considerably between sites. Common characteristics for most sites (except

for Changri Nup) are that wind speeds are highest around December/January and that monsoonal wind speeds are generally higher than during the shoulder seasons.

At each site, we define the onset and recession date of monsoon based on visual inspection of the AWS records, observing the seasonal shift of air temperature, relative humidity, and longwave and shortwave radiation. For this, we define the date after which clear shifts are observable in the variables' regime (Figures A1 to A7).





## 3 Methods

### 3.1 Tethys-Chloris energy balance model

We use the hydrological, snow and ice modules of the Tethys-Chloris (T&C) land surface model (Fatichi et al., 2012; Mastrotheodoros et al., 2020; Paschalis et al., 2018; Botter et al., 2020) to simulate the mass and energy balance of the seven study glaciers. T&C simulates, in a fully distributed manner, the energy and mass budgets of a large range of possible land surfaces,

including vegetated land, bare ground, water, snow and ice. Here, we apply the model at the point scale of the AWS locations to simulate the energy fluxes of the underlying surface and subsurface, which can comprise snow, ice and supraglacial debris cover layers, according to the local and dynamic conditions. The melt and accumulation of ice and snow, and the ice melt under debris are also explicitly simulated. The surface energy balances for the three different possible surfaces are for snow,

$$R_n(T_{sno}) + Q_v(T_{sno}) + Q_{fm}(T_{sno}) - H(T_{sno}) - \lambda E(T_{sno}) - G(T_{sno}) - dQ(T_{sno}) = 0, \tag{1}$$

for debris cover,

$$R_n(T_{deb}) + Q_v(T_{deb}) - H(T_{deb}) - \lambda E(T_{deb}) - G(T_{deb}) = 0, \tag{2}$$

and for ice,

$$R_n(T_{ice}) + Q_v(T_{ice}) - H(T_{ice}) - \lambda E(T_{ice}) - G(T_{ice}) - dQ(T_{ice}) = 0, \tag{3}$$

where $R_n\,[W\,m^{-2}]$ is the net radiation absorbed by the snow/debris/ice surface, $Qv\,[W\,m^{-2}]$ is the energy advected from

precipitation, $Q_{fm}\,[W\,m^{-2}]$ is the energy gained or released by melting or refreezing the frozen or liquid water that is held inside the snow pack, $H\,[W\,m^{-2}]$ is the sensible energy flux and $\lambda E\,[W\,m^{-2}]$ the latent energy flux for any of the surfaces, and $G\,[W\,m^{-2}]$ is the conductive energy flux from the surface to the subsurface. In ice, the conduction of energy is represented in the model down to a depth of $2\,m$ after which it is assumed the ice pack is isothermal. Finally, $dQ\,[W\,m^{-2}]$ is the net energy input to the snow or ice pack. For debris on top of ice, and snow on top of debris or ice, the in-/outgoing fluxes towards/from

the ice are adjusted according to the respective interface type. The sign convention is such that fluxes are positive when directed towards the surface. To close the energy balance, a prognostic temperature for the different surface types ($T_{sno}$, $T_{deb}$, $T_{ice}$) is estimated for each computational element. Iterative numerical methods are used to solve the non-linear energy budget equation until convergence for the ice and snow surface, and the heat diffusion equation for the debris surface, while concurrently computing the mass fluxes resulting from snow and ice melt and sublimation.


### 3.1.1 Radiative fluxes

$R_n$ is calculated as the sum of incoming and outgoing shortwave and longwave fluxes as

$$R_n = SW_\downarrow(1 - \alpha) + LW_\downarrow + LW_\uparrow, \tag{4}$$





where $SW_\downarrow\,[W\,m^{-2}]$ is the incoming shortwave radiation, $\alpha\,[-]$ is the surface albedo, $LW_\downarrow\,[W\,m^{-2}]$ and $LW_\uparrow\,[W\,m^{-2}]$ are the incoming atmospheric and outgoing longwave radiation components, respectively. In this study $\alpha$ is given as an input to the model based on the AWS observations. We prescribe $\alpha$ for all surface types as the daily cumulated albedo, which is the 24 hour sum of $SW_\uparrow$ divided by the sum of $SW_\downarrow$ centred over the time of observation (van den Broeke et al., 2004).

### 3.1.2    Incoming energy with precipitation

For calculating the incoming energy with precipitation, rain is assumed to fall at air temperature $(T_a)$ when positive, with a lower boundary of $0\ °C$. Snow is assumed to fall at negative $T_a$ with an upper boundary of $0\ °C$. Here, $Q_v$ is the energy required to equalize the precipitation temperature with the surface temperature $T_s$ and is therefore calculated as

$$Q_v = c_w\,P_{r,liq}\,\rho_w\,[max(T_a,0)-T_s] + c_i\,P_{r,sno}\,\rho_w\,[min(T_a,0)-T_s], \tag{5}$$

where $c_w = 4196\,[J\,kg^{-1}\,K^{-1}]$ is the specific heat of water, $c_i = 2093\,[J\,kg^{-1}\,K^{-1}]$ the specific heat of ice, $\rho_w = 1000\,[kg\,m^{-3}]$

is the density of water and $P_{r,liq}\,[mm]$, $P_{r,sno}\,[mm]$ are the rain- and snowfall intensities, respectively.

### 3.1.3    Phase changes in the snow pack

The snow pack has a water holding capacity $Sp_{wc}$ described in section 3.2.2. The energy flux gained/released by melting/refreezing the frozen/liquid water that is held inside the snow pack is calculated as:

$$Q_{fm}(t) = \begin{cases} f_{sp}\dfrac{\lambda_f\,\rho_w\,Sp_{wc}(t-dt)}{1000\,dt}, & T_{sno}(t) < 0 \ and \ T_{sno}(t-dt) \geq 0 \\[2mm] f_{sp} - \dfrac{\lambda_f\,\rho_w\,Sp_{wc}(t-dt)}{1000\,dt}, & T_{sno}(t) \geq 0 \ and \ T_{sno}(t-dt) < 0 \end{cases} \tag{6}$$

where $f_{sp} = \frac{5}{SWE}\,[m^{-1}]$ with $max(f_{sp}) = 1$ is the fraction of the snowpack water equivalent ($SWE\,[mm\,w.e.]$) involved in either melting or freezing. This choice was made in order to mimic refreezing in the upper portion of the snowpack, while the snowpack is otherwise represented as a single layer. $\lambda_f = 333700\,[J\,kg^{-1}]$ is the latent energy of melting and freezing of water, $t$ stands for time, $dt\,[s]$ is the timestep, and the unit for $T_{sno}$ is $[°C]$.

### 3.1.4    Turbulent energy fluxes

Over snow, debris and ice surfaces, the sensible energy flux is calculated as

$$H = \rho_a\,C_p\,\frac{(T_s - T_a)}{r_{ah}}, \tag{7}$$

where $T_s\,[°C]$ is the surface temperature (generalised term for $T_{sno}, T_{deb}, T_{ice}$), $C_p = 1005 + [(Ta+23.15)^2]/3364\,[J\,kg^{-1}\,K^{-1}]$ is the specific heat of air at constant pressure, $\rho_a\,[kg\,m^{-3}]$ is the density of air. The aerodynamic resistance $r_{ah}\,[s\,m^{-1}]$, which is also a function of wind speed ($Ws$) is calculated using the simplified solution of the Monin-Obokhov similarity theory

(Mascart et al., 1995; Noilhan and Mahfouf, 1996). The roughness lengths of heat ($z_{0h}\,[m]$) and water vapour ($z_{0w}\,[m]$) used in the calculation of the aerodynamic resistance are equal in T&C ($z_{0h} = z_{0w}$), and ($z_{0h} = z_{0w} = 0.1z_{0m}$). The roughness length



of momentum ($z_{0m}$) is set to 0.001 m for snow and ice surfaces (Brock et al., 2000), while we optimize it against the surface temperature for debris (see section 3.3).

Correct estimates of the latent energy flux due to water phase changes at the surface are important to accurately model glacier melt, especially under moist conditions (Sakai et al., 2004). Phase changes between the water and gas phase and the resulting energy fluxes are considered over all surfaces. The latent energy is limited by the availability of water in the form of ice and snow, or in the case of a debris surface, by the amount of water intercepted (interception storage capacity is set to 2mm). The latent energy flux is estimated from:

$$\lambda E = \lambda_s \frac{\rho_a \left( q_{sat}(T_s) - q_a \right)}{r_{aw}}, \tag{8}$$

where $\lambda_s$ is the latent energy of sublimation defined as $\lambda_s = \lambda + \lambda_f$, with $\lambda = 1000 \left( 2501.3 - 2.361 \, T_a \right) \left[ J \, kg^{-1} \right]$ as the latent energy of vaporisation. $q_{sat}$ is the surface specific humidity at saturation at $T_s$, $q_a$ is the specific humidity of air at the measurement height and $r_{aw}$ the aerodynamic resistance to the vapour flux, which we assume equals $r_{ah}$.

### 3.1.5    Ground energy flux

The definition of the ground energy flux $G \left[ W \, m^{-2} \right]$ differs based on the surface type. In the case of snow, it is equal to the energy transferred from the snowpack to the underlying ice or debris surface, where in the assumption of a slowly changing process, $G$ can be approximated with the temperature difference of the previous time step (t-1), which allows to solve for $G$ outside the numerical iteration to find the snow surface temperature of the current time step:

$$G_{sno}(t) = k_{sno} \frac{T_{sno}(t-1) - T_{deb,ice}(t-1)}{d_{sno}} \tag{9}$$

where $k_{sno} \left[ W \, K^{-1} \, m^{-1} \right]$ is the thermal conductivity of snow and $d_{sno} \left[ m \right]$ is the snow depth. For ice in the absence of snow and debris, it is the energy flux from the ice pack to the underlying surface or to the ice at a depth of $2 \, m$:

$$G_{ice}(t) = k_{ice} \frac{T_{ice}(t-1) - T_{grd}(t-1)}{d_{ice}} \tag{10}$$

where $k_{ice} \left[ W \, K^{-1} \, m^{-1} \right]$ is the thermal conductivity of ice, $T_{grd} \left[ °C \right]$ is the temperature of the underlying ice, and $d_{ice} \left[ m \right]$ is the relevant ice thickness. For debris, which was discretised into eight layers at all debris-covered sites, $G$ is the energy flux

conducted into the debris layers. Its calculation is for a given time $t$ and depth $z$

$$G(z,t) = -k_d \frac{\partial T_{deb}(z,t)}{\partial z_d}, \tag{11}$$

where $k_d \left[ W \, K^{-1} \, m^{-1} \right]$ is the debris thermal conductivity (see section 3.3) and $T_{deb}(z,t) \left[ °C \right]$ is the debris temperature at time $t$ and depth $z$. $G(z,t)$ can be included in the heat diffusion equation as such:

$$cv_s \frac{\partial T_{deb}(z,t)}{\partial t} = \frac{\partial}{\partial z_d} \left( -G(z,t) \right), \tag{12}$$





where $cv_d$ is the debris heat capacity. Under the assumption of homogeneous debris layers, $\kappa\,[m^2\,s^{-1}]$ as the debris heat diffusivity replaces the term $\frac{k_d}{cv_s}$ and equation (12) can be written as:

$$\frac{\partial T_{deb}(z,t)}{\partial t} = \kappa\frac{\partial^2 T_{deb}(z,t)}{\partial z^2}, \tag{13}$$

The heat diffusion equation (13) is solved using iterative numerical methods. This way, the debris temperature profile $T_{deb}(z,t)$ is solved together with $G(z,t)$ at any depth and time. The conductive energy flux at the base of the debris is used to
heat the ice and to calculate ice melt once above the melting point.

### 3.2   Mass balance in T&C

#### 3.2.1   Precipitation partition

Input precipitation is required to be partitioned into solid $Pr_{sno}$ and liquid $Pr_{liq}$ precipitation, because of the differing impacts of snow and rain on the energy and mass balance. For this study, the precipitation partition method described by Ding et al.
(2014) was implemented into T&C. This parameterisation has been developed specifically for High Mountain Asia based on a large dataset of rain, sleet and snow observations, and does not require recalibration. It determines the precipitation partition based on the wet-bulb temperature, station elevation and relative humidity and allows for sleet events, as a mixture between liquid and solid precipitation. Ding et al. (2014) found the wet-bulb ($T_{wb}$) to be a better predictor than $T_a$ of the precipitation type, that the temperature threshold between snow and rain increases at higher elevations, and that the probability of sleet is
reduced in conditions of low relative humidity.

#### 3.2.2   Water content of the snow, ice and debris layers

The water content of ice is approximated with a linear reservoir model. The liquid water outflow is proportional to the ice pack water content $Ip_{wc}\,[mm\,w.e.]$, which is initiated when $Ip_{wc}$ exceeds a threshold capacity, prescribed as 1% of the ice water equivalent ($IWE\,[mm\,w.e.]$). The $Ip_{wc}$ is the sum of ice melt and liquid precipitation, minus the water released from the ice
pack. The water released is the sum of the ice pack excess water content plus the outflow from the linear reservoir, given as $I_{out} = Ip_{wc}/K_{ice}$ , where $K_{ice}$ is the reservoir constant which is proportional to the ice pack water equivalent. Unlike within snow packs, $Q_{fm}$ is not accounted for within the ice pack, since water is presumed to be evacuated quickly from the ice due to runoff without refreezing.

The water content of the snow pack $Sp_{wc}\,[mm\,w.e.]$ is approximated using a bucket model, in which outflow of water from
the snow pack occurs when the maximum holding capacity of the snow pack is exceeded. Following the method of Bélair et al. (2003), the maximum holding capacity of the snow pack is based on $SWE$, a holding capacity coefficient and $rho_{sno}$. Snowmelt plus liquid precipitation, minus the water released from the snow pack gives the current $Sp_{wc}$. If $T_{sno}$ is greater than $0°C$ then the snow pack water content is assumed to be liquid, whereas otherwise it is assumed frozen.

For supraglacial debris, both observations and methods for modelling its water content are lacking. We thus use a simplified





270 scheme for accounting for moisture at the surface of the debris, in order to mimic the drying process of the debris surface: we assume debris to have a dynamic interception storage, which can hold a maximum of $2\,mm$ water at all debris-covered sites and can be refilled by snowmelt or liquid precipitation. The evaporative flux from the debris and the latent energy flux of evaporation is therefore limited by this interception storage.

### 3.2.3  Glacier mass balance


The mass balance calculation of snow and ice is rather similar, so they will be described together here. Calculations are performed for snow if there is snow precipitation during a timestep or the modelled $SWE$ at the surface is greater than zero. Net input of energy to the snow or ice pack will increase its temperature, and after the temperature has been raised to the melting point, additional energy inputs will result in melt. The change in the average temperature of the ice or snowpack $dT$ is
280 controlled using:

$$dT = \frac{dQ\,dt}{c_i\,\rho_w\,WE_b}1000,\qquad(14)$$

Where $dt$ is the time step $[h]$ and $WE_b\,[mm\,w.e.]$ is $IWE$ or $SWE$ before melting and limited to a maximum of $2000\,mm$, assumed to be the water equivalent mass exchanging energy with the surface. Energy inputs into an iso-thermal ice/snow pack result in melt $M\,[mm\,w.e.]$ as

285 $$M = \frac{dQ\,dt}{\lambda_f\,\rho_w}1000\qquad(15)$$

The water equivalent mass of the snow/ice pack after melting $WE(t)\,[mm\,w.e.]$ is updated conserving the mass balance following:

$$WE(t) = WE(t-dt) + Pr_{sno}(t) - E(t)\,dt - M(t),\qquad(16)$$

Here $E = \lambda E/\lambda s\,[mm]$ is the sublimation from ice and snow. The snow density is assumed to be constant with depth and cal-
290 culations are performed assuming one single snow pack layer. The snow density evolves over time using the method proposed by Verseghy (1991) and improved by Bélair et al. (2003). In this parameterisation the snow density increases exponentially over time due to gravitational settling and is updated when fresh snow is added to the snowpack. Two parameters are required in this scheme, $\rho_{sno}^{M1}$ and $\rho_{sno}^{M2}\,[kg\,m^{-3}]$, which represent the maximum snow density under melting and freezing conditions, respectively. The depth of the ice pack can be increased through the formation of ice from the snow pack (ice accumulation),
295 which is prescribed to occur if the snow density increases to greater than $500\,kg\,m^{-3}$ (a density associated with the firn to ice transition) and at a rate of $0.037\,mm\,h^{-1}$ (Cuffey and Paterson, 2010). The density of ice is assumed constant with depth and equal to $916.2\,kg\,m^{-3}$.

### 3.3  Debris parameters

A major challenge in physically based mass-balance modelling of debris-covered glaciers is the assignment of appropriate
300 debris properties. Besides the debris thickness, which was measured at the AWS location, the thermal conductivity $k_d$, the





roughness length $z_0$ of the debris surface, the surface emissivity $\epsilon_d$, the debris volumetric heat capacity $cv_d$ and and the debris density $\rho_d$ have to be assigned. While the latter three can be quantified using literature values, there is more uncertainty about $k_d$ and $z_0$, two highly variable quantities that are difficult to measure in the field, but which EB models are highly sensitive to. We thus choose to optimize them, since our primary requirement is an accurate representation of the energy and mass balance:

(1) in a first step, we optimize $k_d$ simulating only the conduction of energy through the debris during snow free conditions, with the $LW_\uparrow$-derived surface temperature $T_{s,LW}$ as an input, the ice melt as the target variable and the Nash-Sutcliffe Efficieny $NSE\,[-]$ as performance metric. (2) Next, we run the whole energy balance model and optimize $z_0$ for snow-free conditions, with all required meteorological inputs, and the optimal $k_d$ from step (1), while comparing modelled $T_s$ against $T_{s,LW}$, using $NSE$ as performance metric. The resulting parameters are given in Table 3.

**3.4 Uncertainty estimation**

We calculate the uncertainty associated with all energy and mass balance components by performing $10^3$ Monte Carlo simulations for each study site at the AWS location. We vary three debris parameters ($k_d$,$z_{0m}$,$\epsilon_d$), debris thickness $h_d$, as well as six measured model input variables: air temperature $T_a$, the vapor pressure at reference height $ea\,[-]$, $SW_\uparrow$, $SW_\downarrow$, $LW_\downarrow$, the total precipitation before partition $Pr$, and the wind speed $Ws$. Measured outgoing shortwave radiation $SW_\uparrow$ was included into the

Monte Carlo set, as it determines our input $\alpha$, as discussed in Section 3.1.1. While the parameter uncertainty range was defined based on previously published values for debris (e.g. Yang et al., 2017; Rounce et al., 2015; Evatt et al., 2015; Reid and Brock, 2010; Nicholson and Benn, 2006; Rowan et al., 2020; Lejeune et al., 2013; Collier et al., 2015; McCarthy, 2018), the debris thickness measurement uncertainty was given with a range of $1cm$ and the range for the meteorological inputs was set based on the respective sensor uncertainties (see Table 4). All uncertainties were equally distributed around the standard parameter

values and observations. Uncertainties are given as one standard deviation of the error of the Monte Carlo runs against the standard run.

**Table 3.** Debris parameter values for each site derived from the two-step optimization procedure.

|  | Lirung | Langtang | Changri Nup | 24K | Hailuogou |
|---|---|---|---|---|---|
| $k_d\,[W\,m^{-1}\,K^{-1}]$ | 1.09 | 1.65 | 1.77 | 1.45 | 0.72 |
| $z_{0m}\,[m]$ | 0.7 | 0.38 | 0.11 | 0.15 | 0.027 |

**4 Results**

**4.1 Modelled mass balance**

The model accurately reproduces the measured surface height change (ablation and accumulation) at both debris-covered and

clean-ice glaciers (Figure 4). The maximum uncertainties associated with each model output ranges from $\pm4\%$ (Parlung No.4, Figure 4f) to $\pm15\%$ (Yala, Figure 4c). Where Ultrasonic Depth Gauge (UDG) records are available (Lirung, Langtang, Yala,





**Table 4.** Uncertainty ranges of parameters and input variables used for Monte Carlo runs. Where units are indicated with [-], the parameter or variable was perturbed by the fractional value shown.

| Parameter/ Variable | Range | Parameter/ Variable | Range |
|---|---|---|---|
| $k_d\,[-]$ | $\pm 0.1$ | $SW_\downarrow\,[-]$ | $\pm 0.03$ |
| $z0\,[-]$ | $\pm 0.1$ | $SW_\uparrow\,[-]$ | $\pm 0.03$ |
| $\epsilon_d\,[-]$ | $\pm 0.05$ | $LW_\downarrow\,[-]$ | $\pm 0.03$ |
| $h_d\,[mm]$ | $\pm 5$ | $Pr\,[-]$ | $\pm 0.15$ |
| $Ta\,[°C]$ | $\pm 0.2$ | $W_s\,[m/s]$ | $\pm 0.3$ |
| $ea\,[-]$ | $\pm 0.02$ | | |

Changri Nup), the deviations of the simulations from the observations stay within the uncertainty range (Figure 4a-d). We decided to not consider the UDG record from Changri Nup after a large August snowfall, as variables describing the surface state (e.g. $\alpha$, $LW_\uparrow$) following this event indicate a discontinuous snow cover at the AWS location, while the UDG, which is

some meters away from the AWS, shows continuous snow cover with depths at the order of tens of centimeters. This discrepancy was also confirmed by observation of the site from October 2016. It was thus not possible to match the UDG record with our model for the late ablation period on Changri Nup, but our model closely reproduces observed surface height change for the pre-monsoon and early monsoon (Figure 4d), when AWS and UDG observations agree in terms of surface state. The deviation to measured melt is larger than the uncertainty range at 24K and Parlung No.4 for two and one individual stake

readings, respectively, but the overall agreement is very good also at these sites (Figure 4e,f). For Parlung No.4 there are no stake measurements available before July 21 due to the long-lasting snow cover.

Mass losses over the ablation seasons (combining ice melt, snow melt and sublimation) show considerable variability between study sites and over time at each site: At all sites, ice melt is the dominant mass loss component, accounting for 65.8% (Changri Nup, Figure 5c) to 95.4% (Hailuogou, Figure 5g) of the total mass losses. It increases initially during premonsoon at all sites,

then tends to plateau at Lirung and Langtang. Snow melt is a smaller component of mass loss, accounting for 0.1% (24K, Figure 5e) to 33.1% (Hailuogou, Figure 5g) of the total mass losses. However, it is an important control of both amounts and patterns of ice melt, as ice melt rates are suppressed during periods of snow cover (Figure 5). A longlasting pre-monsoonal snowpack can delay the onset of ice melt, as at Parlung No.4, where ice melt is delayed until the end of June (Figure 5f). Similarly, intermittent snow cover protects the ice from melting at the two highest sites (Yala and Changri Nup) during the summer

months (Figure 5c and d). Sublimation from ice and snow represents a very small share of the total mass losses, and ranges from 0.01% (Lirung, Figure 5a) to 1.2% (Changri Nup, Figure 5d). It mostly occurs under dry conditions during premonsoon at the highest sites (Changri Nup, Yala). Debris cover is clearly another important control of total mass losses: The presence

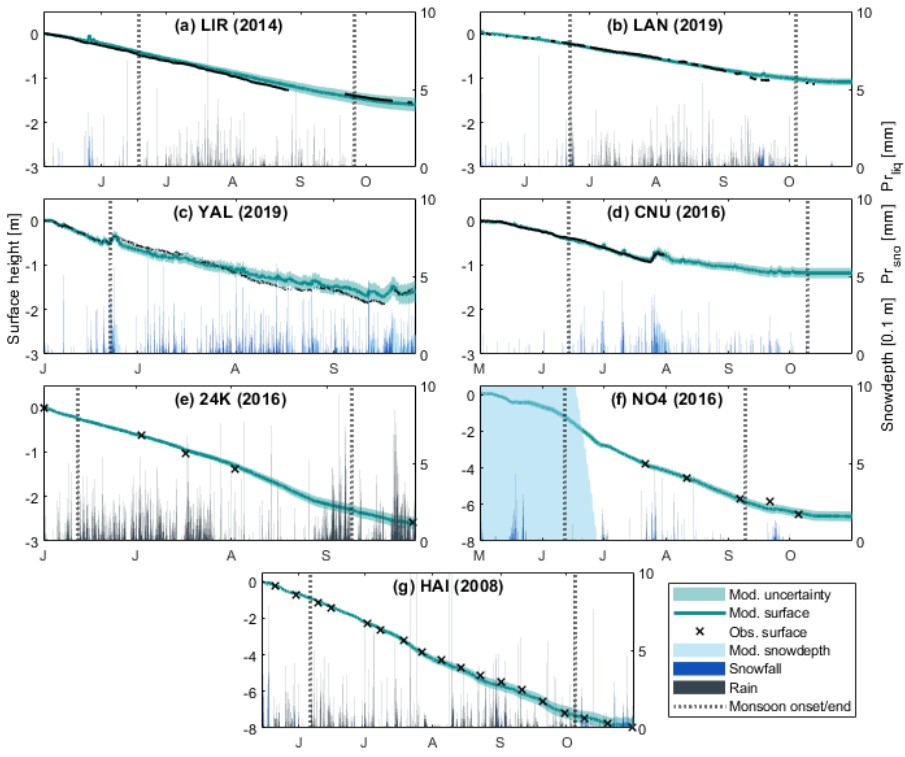

**Figure 4. (a)-(g)** Measured vs. modelled surface change at all study sites, including ice melt, snow melt, sublimation, precipitation phase and snow depth. Measured melt is either from ablation stakes (black circles) or Ultrasonic Depth Gauges (black lines). Vertical dotted lines indicate monsoon onset and end.

of debris at least $10\,cm$ thick (Changri Nup, Lirung, Langtang and 24K) causes average melt rates to be comparatively low (8.2, 8.5, 6.1 and $19.4\,mm\,d^{-1}$, Figure 5a,b,d and e), while at the 'dirty-ice' site (Hailuogou, with thin and patchy debris, approximated to be 1 cm), the melt rate is higher ($42.7\,mm\,d^{-1}$, Figure 5g) over the simulation period, peaking in mid-July. Relatively high average melt rates are also reached at the clean ice sites Parlung No.4 ($34.6\,mm\,d^{-1}$, Figure 5f) and Yala ($16.6\,mm\,d^{-1}$, Figure 5c), despite the high altitude of the latter.

### 4.2 Modelled energy balance

The main energy source on all glaciers and during all seasons is $SW_{net}$ (Figure 6). At all sites and over the entire modelling period, $LW_{out}$ is the largest energy sink, but it remains comparably stable in magnitude between the seasons (Figure A8). The turbulent heat fluxes ($H$ and $LE$), and $dQ$ ($G$ on debris) act as energy sinks on average at all sites. $G$ can however act in both ways, as an energy source when warming the glacier, and as an energy sink when warming the air. In our definition $G$ sums





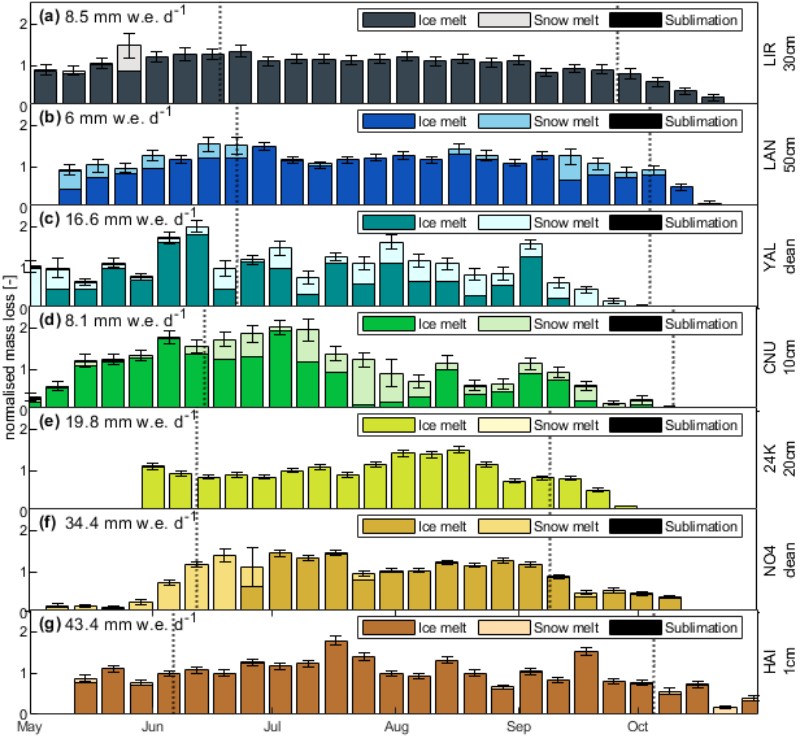

**Figure 5. (a)-(g)** Melt rates of ice and snow (stacked) as weekly averages at each site. Vertical dotted lines indicate monsoon onset and end. Error bars depict the uncertainty (standard deviation) of the estimates. Melt rates are normalized to the mean of the ice melt over the entire period (value in the upper left of each panel).

up all types of conductive energy fluxes in the snow-debris-ice column. Part of the energy is used for heating the snow, ice or debris-ice surface layer until melt occurs ($dG$, Table 5). The relative contributions of the individual sinks strongly depends on the surface type: $LW_{net}$ is an important sink at the clean-ice sites, while the turbulent fluxes remain small there. On the debris-covered glaciers with debris >10 cm thick, where $LW_{net}$ plays a relatively small role, the turbulent fluxes act as major energy sink. Where debris is thin, they instead act as additional energy sources.

In the following, we consider how energy fluxes change when moving from pre-monsoon to monsoon. Note that the sign of this change depends on the sign of the original flux, which is negative when the flux acts as an energy sink (energy away from the surface) and positive when the flux acts as a source (energy towards the surface). As an example, a negative change in $LE$, when acting as a sink, means an increase in its magnitude.

The seasonal imprint of the monsoon on the energy balance is very clear at all study sites on average and diurnally: There
is a strong reduction in the magnitude of $SW_{\downarrow}$ during monsoon across sites (with decreases ranging between $-41.8$ and $-135\,W\,m^{-2}$ at the seven sites, Figure 7 and Table 5), which however remains the main energy source. $SW_{\uparrow}$, which is a sink controlled by surface albedo, follows these changes (ranging from $+5.4$ to $+164.8\,W\,m^{-2}$ between sites) with the exception





of Changri Nup, where $SW_\uparrow$ increases in magnitude (by $-12.1\,W\,m^{-2}$, Table 5) as a consequence of ephemeral snow cover (Figure 4). $LW_\downarrow$ on the other hand increases at all sites (by $+15.7$ to $+57.0\,W\,m^{-2}$), and $LW_\uparrow$ slightly increases in magnitude
as well (by $-1.0$ to $-13.3\,W\,m^{-2}$) (Figure 7 and Table 5).

At the debris-covered sites, the reduction in $R_{net}$ from pre-monsoon to monsoon is balanced by a reduction in the magnitude of $H$ (with changes of $+15.6$ to $+67.7\,W\,m^{-2}$ between sites), which acts as a major energy sink during both sub-seasons (Figure 6, 7 and Table 5). $LE$, which plays a small role as a sink compared to $H$ during pre-monsoon, increases in magnitude at the debris-covered sites (with changes ranging from $-2.7$ to $-24.4\,W\,m^{-2}$, Figure 7 and Table 5), and acts as a considerable heat
sink during the monsoon. As a result of these contrasting changes, $dQ$ remains fairly similar between the pre-monsoon and monsoon at the debris covered sites (with changes ranging between $+1.3$ and $-3.3\,W\,m^{-2}$) (Figure 7a,c,e,g and Table 5). This is partly a consequence of a compensation between increased/reduced melt rates before/after noon (Figure A10). An exception is the thin-debris site Hailuogou, where $dQ$ increases in magnitude (by $-26.2\,W\,m^{-2}$), mostly as a result of warmer nights (Figure A10).

At the clean-ice sites, $LW_\uparrow$ is initially larger than $LW_\downarrow$ in the pre-monsoon, but the two become almost equal while adjusting to monsoon conditions, causing the $LW_{net}$ to be close to zero ($LW_\downarrow$: $+40.7$ to $+57.0\,W\,m^{-2}$, and $LW_\uparrow$: $-3.1$ to $-11.7\,W\,m^{-2}$), Table 5 and Figure 6c and f). $LE$ switches sign when moving from pre-monsoon to monsoon, becoming a small condensation energy flux (Yala: 0.1, Parlung No.4: $3.8\,W\,m^{-2}$). So does $H$ at Yala, becoming a small heat source instead of a heat sink ($0.5\,W\,m^{-2}$, Table 5), while at Parlung No.4, $H$ is a heat source across the seasons (pre-monsoon: $4.9\,W\,m^{-2}$, monsoon:
$12.7\,W\,m^{-2}$, Table 5). The turbulent fluxes, however, remain comparably small in magnitude at the clean ice sites (Figure 6c and f). $dQ$ increases in magnitude at Parlung No.4 from pre-monsoon to monsoon (by $-131.0\,W\,m^{-2}$) and decreases in magnitude at Yala (by $-10.2\,W\,m^{-2}$) (Figure 7b and d). Both changes are consequences of snow cover, prevalent during pre-monsoon at Parlung No.4 and during monsoon at Yala (Figure 4c and f).

Hailuogou, our 'dirty ice glacier', behaves entirely differently compared to both debris-covered and clean sites, with the turbu-
lent energy fluxes increasing in magnitude to become considerable heat sources during monsoon ($H$: $+15.6\,W\,m^{-2}$ and $LE$:, $+24.5\,W\,m^{-2}$), driving a pronounced increase in the energy available for melt ($dQ$: $-26.2\,W\,m^{-2}$, Figure 6, 7 and Table 5).

### 4.3   Turbulent fluxes at debris-covered sites and their controls

The turbulent fluxes $LE$ and $H$ are important heat sinks on debris-covered glaciers and heat sources on the dirty-ice glacier,
while they remain small on the clean-ice glaciers (Figure 6). Their magnitude largely determines the energy that is conducted into the debris and used for ice melt. We observe a decrease in the magnitude of $H$ and an increase in the magnitude of $LE$ going from pre-monsoon to monsoon at the debris-covered glaciers, while both increase as sources of energy at the dirty ice glacier (Figure 7).

The predictive power of the temperature gradient between surface and air $gT\,[°C^{-1}]$ and $Ws$ as well as their combination for
determining $H$ and $LE$ were assessed using a univariate polynomial regression model for the single predictors and a multiple polynomial regression model using both variables, and both models had 2 degrees of freedom. $H$ is largely controlled by the



temperature gradient between surface and air ($g_T$) on debris-covered glaciers, which explains between 75 and 99% of the variability of $H$ (Figure A9), and $g_T$ decreases during monsoonal conditions by $-0.05$ to $-1.44\,^{\circ}C\,m^{-1}$ (Table 6). Here it is clear that a smaller temperature gradient between surface and air during the monsoon reduces the magnitude of $H$ as a

sink. Wind emerges as the most important control of $H$ at the dirty-ice glacier, explaining up to 89% of variability (Figure A9). The mean magnitude of $Ws$ increases at that site from $1.23\,m\,s^{-1}$ in pre-monsoon to $2.10\,m\,s^{-1}$ in monsoon (Table 6). A cold surface in combination with a wind-enhanced turbulence and fast turnover of warm air mass results in $H$ becoming a potent heat source on the dirty-ice glacier. From a physical perspective, the same holds for clean-ice glaciers experiencing simultaneously high wind speeds and high air temperatures. This is, however, not observed on Yala and Parlung No.4, where

air temperatures stay comparably low and average wind speed decreases slightly in monsoon compared to pre-monsoon (Table 6).

Neither $RH$, $g_T$, or $Ws$ on their own explain the variability of $LE$ across sites (Figure A9). $LE$ however increases consistently from pre-monsoon to monsoon together with an increase in the duration of moisture availability at the surface of debris-covered glaciers (by 21 to 63%, Table 6). In fact, evaporation and the related $LE$ release tends to be water-limited during pre-monsoon,

but energy-limited during monsoon (Figure 8). This implies that the availability of moisture is driving the increase of $LE$ from pre-monsoon to monsoon. However, on the dirty-ice glacier, $LE$ acts as a heat source across sub-seasons and switches sign during monsoon at the clean-ice sites. This happens when $T_a$ is consistently higher than $T_s$, causing condensation when the warm air touches the cold glacier-, or thin debris-surface.



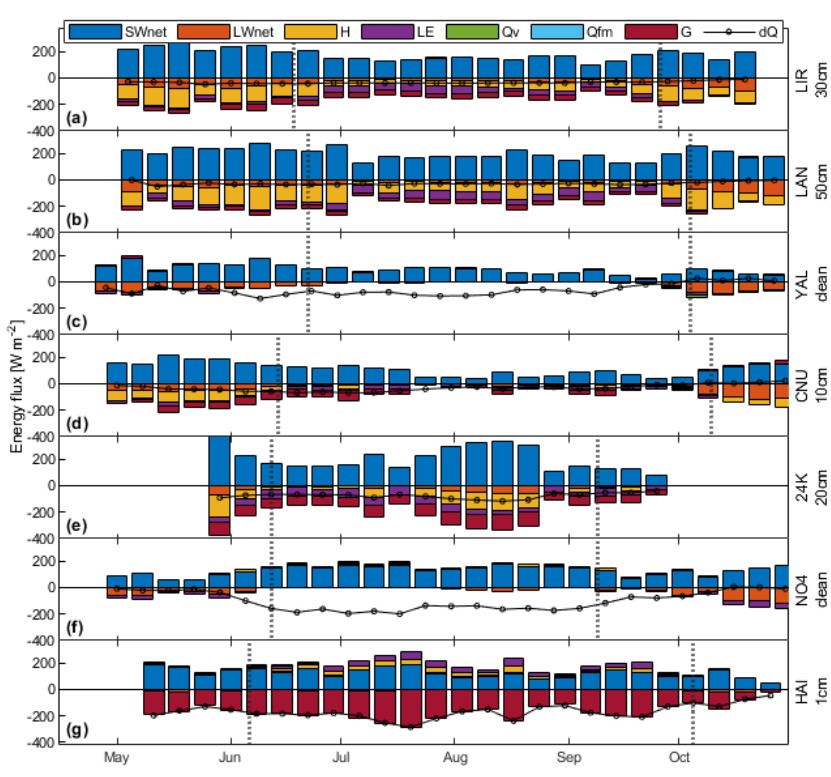

**Figure 6. (a)-(g)** Stacked energy fluxes weekly averages at each site, depicting the components $SW_{net}$, $LW_{net}$, $H$, $LE$, $Q_v$, $Q_{fm}$, $G$ and $dQ$. Energy fluxes are negative fluxes when directed away from the surface and positive when directed towards the surface.




**Table 5.** Mean energy balance components at each site and for pre-monsoon (pre), monsoon (mon), and post-monsoon (post) periods, as well as flux magnitude changes from pre-monsoon to monsoon, and monsoon to post-monsoon. All values are in $W\,m^{-2}$.

| Site | | SWin pre | SWin mon | SWin post | SWout pre | SWout mon | SWout post | LWin pre | LWin mon | LWin post | LWout pre | LWout mon | LWout post | LE pre | LE mon | LE post |
|---|---|---|---|---|---|---|---|---|---|---|---|---|---|---|---|---|
| LIR | mean | 277.1 | 170.0 | 224.1 | -39.8 | -21.1 | -41.3 | 293.3 | 341.2 | 264.0 | -358.1 | -364.8 | -340.4 | -16.0 | -40.4 | -1.9 |
| | Δ | | -107.1 | 54.1 | | 18.7 | -20.2 | | 47.9 | -77.2 | | -6.7 | 24.4 | | -24.4 | 38.5 |
| LAN | mean | 295.6 | 208.1 | 262.4 | -55.9 | -30.1 | -48.6 | 292.3 | 334.7 | 238.6 | -339.3 | -352.8 | -326.1 | -26.4 | -49.9 | -5.8 |
| | Δ | | -87.5 | 54.3 | | 25.7 | -18.4 | | 42.3 | -96.0 | | -13.5 | 26.7 | | -23.5 | 44.1 |
| YAL | mean | 307.7 | 172.7 | 271.8 | -169.4 | -94.4 | -195.0 | 248.5 | 305.5 | 212.3 | -299.8 | -311.6 | -287.1 | -13.7 | 0.1 | -12.2 |
| | Δ | | -135.0 | 99.1 | | 75.0 | -100.6 | | 57.0 | -93.2 | | -11.7 | 24.4 | | 13.7 | -12.3 |
| CNU | mean | 237.1 | 154.7 | 258.1 | -60.6 | -72.7 | -126.0 | 268.8 | 310.2 | 196.4 | -318.7 | -319.7 | -300.4 | -15.7 | -18.1 | -3.1 |
| | Δ | | -82.4 | 103.4 | | -12.1 | -53.3 | | 41.5 | -113.8 | | -1.0 | 19.3 | | -2.5 | 15.0 |
| 24K | mean | 296.6 | 219.7 | 140.9 | -18.5 | -13.0 | -8.3 | 324.6 | 349.3 | 336.6 | -369.9 | -371.3 | -351.0 | -50.6 | -52.7 | -40.0 |
| | Δ | | -76.9 | -78.8 | | 5.4 | 4.7 | | 24.7 | -12.7 | | -1.3 | 20.3 | | -2.1 | 12.7 |
| NO4 | mean | 308.5 | 209.1 | 197.3 | -219.6 | -54.8 | -81.5 | 267.5 | 308.2 | 261.5 | -310.3 | -313.4 | -310.5 | -17.6 | 3.6 | -18.5 |
| | Δ | | -99.5 | -11.8 | | 164.8 | -26.7 | | 40.7 | -46.7 | | -3.1 | 2.9 | | 21.2 | -22.1 |
| HAI | mean | 178.2 | 136.4 | 105.8 | -25.2 | -10.8 | -28.4 | 314.6 | 330.3 | 273.1 | -324.5 | -327.9 | -309.1 | 5.4 | 31.6 | -4.8 |
| | Δ | | -41.8 | -30.6 | | 14.4 | -17.6 | | 15.7 | -57.2 | | -3.4 | 18.8 | | 26.2 | -36.4 |

| Site | | H pre | H mon | H post | G pre | G mon | G post | Qv pre | Qv mon | Qv post | dQ pre | dQ mon | dQ post | dG pre | dG mon | dG post |
|---|---|---|---|---|---|---|---|---|---|---|---|---|---|---|---|---|
| LIR | mean | -116.7 | -48.4 | -86.5 | -33.7 | -36.4 | -18.7 | 0.0 | 0.0 | 0.0 | -37.5 | -36.5 | -19.0 | 5.6 | 0.0 | 16.6 |
| | Δ | | 68.3 | -38.0 | | -2.7 | 17.8 | | 0.0 | 0.0 | | 1.0 | 17.5 | | -5.6 | 16.6 |
| LAN | mean | -126.7 | -77.4 | -111.4 | -22.5 | -26.2 | -7.8 | 0.0 | -0.3 | 0.0 | -26.9 | -28.7 | -10.4 | 2.2 | 0.0 | 27.2 |
| | Δ | | 49.3 | -34.1 | | -3.7 | 18.4 | | -0.3 | 0.3 | | -1.8 | 18.4 | | -2.2 | 27.2 |
| YAL | mean | -2.3 | 0.5 | 1.4 | 4.7 | 0.9 | 0.0 | 0.0 | -0.2 | -4.1 | -74.8 | -64.6 | -0.3 | 4.7 | 0.9 | 0.0 |
| | Δ | | 2.8 | 0.9 | | -3.8 | -0.9 | | -0.2 | -3.9 | | 10.2 | 64.3 | | -3.8 | -0.9 |
| CNU | mean | -67.4 | -8.7 | -36.1 | -39.2 | -29.1 | 10.4 | 0.0 | 0.0 | 0.0 | -34.6 | -34.1 | -0.2 | 17.8 | 2.1 | 10.7 |
| | Δ | | 58.7 | -27.4 | | 10.1 | 39.5 | | 0.0 | 0.0 | | 0.5 | 33.9 | | -15.7 | 8.6 |
| 24K | mean | -99.8 | -50.8 | -24.4 | -81.8 | -81.4 | -53.1 | -0.3 | 0.2 | 0.6 | -79.5 | -81.6 | -53.7 | 0.3 | 0.0 | 0.0 |
| | Δ | | 49.0 | 26.4 | | 0.4 | 28.4 | | 0.6 | 0.4 | | -2.1 | 27.9 | | -0.3 | 0.0 |
| NO4 | mean | 4.7 | 12.3 | 5.8 | 0.2 | 0.1 | 0.1 | 0.2 | 0.2 | 0.0 | -32.3 | -162.6 | -54.1 | 0.2 | 0.1 | 0.0 |
| | Δ | | 7.6 | -6.6 | | -0.1 | -0.1 | | 0.0 | -0.2 | | -130.4 | 108.5 | | -0.1 | -0.1 |
| HAI | mean | 9.1 | 25.7 | -9.9 | -150.4 | -186.4 | -16.5 | 1.2 | 1.9 | 0.1 | -158.1 | -186.8 | -36.9 | 2.1 | 0.5 | 18.8 |
| | Δ | | 16.6 | -35.5 | | -35.9 | 169.9 | | 0.7 | -1.8 | | -28.7 | 149.9 | | -1.6 | 18.3 |

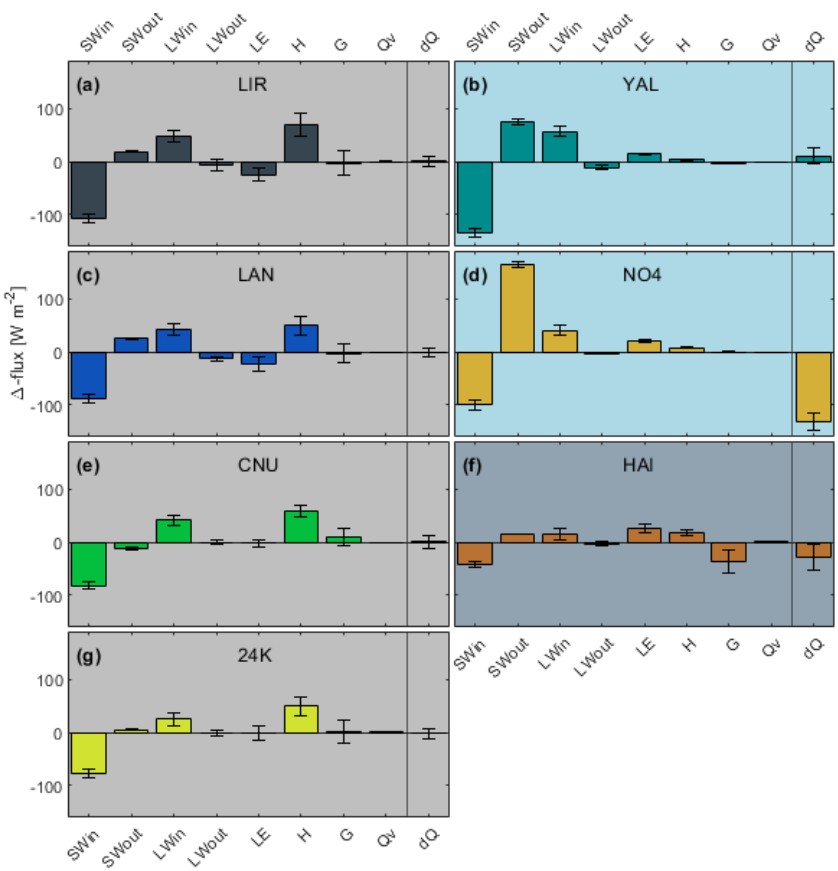

**Figure 7. (a)-(g)** Differences in energy balance components from pre-monsoon to monsoon at each site including their uncertainties (error bars). The direction of change is to be considered relative to the sign of the original flux (x-axis). For example, a positive change in a negative flux means a reduction in the flux, and can also lead to a change in sign. Background indicates the surface type of the site: grey indicates debris-covered, light blue indicates clean-ice sites, and grey-blue indicates thin-debris.

## 5 Discussion

### 5.1 Monsoon effects

Monsoon impacts on glaciers are complex and spatially variable along the Himalayan arc. The varying strength of the monsoons reflect in the distinct meteorological seasonality between our study sites (Figure 3). The ablation period occurs between April and November at all sites, and all sites are affected by the Indian and East Asian summer monsoons during this period (Figures A1 to A7). On average, 70 to 85% of precipitation arrives during the summer months (June-September) at our Central Himalayan sites (Lirung, Lantang, Yala and Changri Nup, Figure 3a-d) in contrast to 40 to 55% at the eastern sites (24K, Parlung No.4 and Hailuogou, Figure 3e-g).

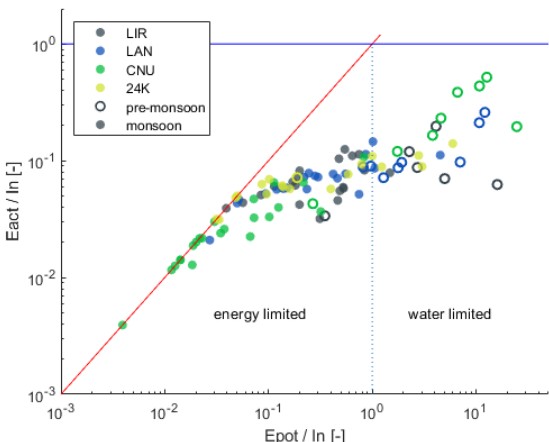

**Figure 8.** Budyko diagram with the 5-day mean potential evaporation rate during snow-free conditions ($Epot$) relative to the mean available intercepted water ($In$) on the x-axis, and the actual evaporation rate during snow-free conditions ($Eact$) relative to $In$ on the y-axis. Only debris-covered glaciers, where $LE$ is a sink term on average, are shown.

**Table 6.** Mean cloud-cover fraction ($ccf$), relative humidity ($RH$), temperature gradient between surface and air ($g_T$), wind speed ($Ws$) and the percentage of time during which the debris is modelled to hold intercepted water ($In$) for each site and season, also indicating percent changes between the sub-seasons.

| | | $ccf\,[-]$ | | | $RH\,[\%]$ | | | $g_T\,[°C\,m^{-1}]$ | | | $Ws\,[m\,s^{-1}]$ | | | $In\,[\%]$ | | |
| --- | --- | --- | --- | --- | --- | --- | --- | --- | --- | --- | --- | --- | --- | --- | --- | --- |
| | | pre | mon | post | pre | mon | post | pre | mon | post | pre | mon | post | pre | mon | post |
| **LIR** | mean | 0.76 | 0.91 | 0.74 | 68.1 | 90.8 | 67.1 | 1.19 | 0.78 | 0.89 | 0.47 | 0.27 | 0.52 | 40.1 | 74.0 | 26.4 |
| | Δ | | 0.14 | -0.16 | | 22.6 | -23.6 | | -0.40 | 0.10 | | -0.19 | 0.24 | | 33.9 | -47.6 |
| **LAN** | mean | 0.76 | 0.86 | 0.62 | 80.7 | 96.9 | 56.3 | 1.02 | 0.97 | 0.95 | 1.79 | 1.10 | 1.27 | 38.8 | 75.3 | 9.5 |
| | Δ | | 0.10 | -0.24 | | 16.2 | -40.6 | | -0.05 | -0.02 | | -0.68 | 0.17 | | 36.5 | -65.8 |
| **YAL** | mean | 0.65 | 0.87 | 0.55 | 69.8 | 93.0 | 39.4 | -0.36 | -0.96 | -0.89 | 1.74 | 1.00 | 1.68 | - | - | - |
| | Δ | | 0.21 | -0.32 | | 23.2 | -53.6 | | -0.59 | 0.07 | | -0.74 | 0.67 | | - | - |
| **CNU** | mean | 0.78 | 0.89 | 0.58 | 71.2 | 89.2 | 39.3 | 1.69 | 0.25 | 0.89 | 1.88 | 1.09 | 2.48 | 16.2 | 79.3 | 5.3 |
| | Δ | | 0.11 | -0.31 | | 18.1 | -50.0 | | -1.44 | 0.64 | | -0.79 | 1.39 | | 63.1 | -74.0 |
| **24K** | mean | 0.70 | 0.82 | 0.91 | 73.1 | 80.6 | 81.2 | 1.86 | 0.73 | 0.18 | 1.33 | 1.56 | 1.35 | 56.8 | 79.1 | 84.3 |
| | Δ | | 0.12 | 0.08 | | 7.4 | 0.7 | | -1.13 | -0.55 | | 0.22 | -0.21 | | 22.3 | 5.2 |
| **NO4** | mean | 0.72 | 0.87 | 0.79 | 65.7 | 81.3 | 73.1 | -0.78 | -2.01 | -0.62 | 2.96 | 2.67 | 3.23 | - | - | - |
| | Δ | | 0.14 | -0.07 | | 15.7 | -8.3 | | -1.22 | 1.39 | | -0.28 | 0.56 | | - | - |
| **HAI** | mean | 0.88 | 0.93 | 0.91 | 81.3 | 92.3 | 90.6 | -2.08 | -2.61 | 0.38 | 1.23 | 2.15 | 0.93 | 99.8 | 100.0 | 75.6 |
| | Δ | | 0.05 | -0.01 | | 10.9 | -1.6 | | -0.53 | 2.99 | | 0.91 | -1.22 | | 0.2 | -24.4 |

### 5.1.1 All glaciers

Overcast conditions caused by monsoon increase $LW_\downarrow$ and with that the magnitude of $T_a$ at all sites (Figures 6 and 7). $SW_\downarrow$ on the other hand is reduced at all glacier surfaces due to the reflection and scattering of persistent, heavy clouds. While the


changes of these two primary sources of energy partly counteract each other in terms of energy supply to glacier surfaces during monsoon, we find that net incoming radiation decreases in magnitude everywhere. Another major control on the energy and mass balance of all glaciers are the snowcover dynamics, which in turn are driven by the precipitation seasonality and the partition of precipitation into rain and snow. For example, in the case of Parlung No.4, the onset of glacier melt was delayed until well after monsoon onset, until all snow had disappeared (Figure 5). After snow has melted out, ephemeral snowcover from monsoonal precipitation increases surface albedo and raises $SW_\uparrow$, protecting the ice and suppressing melt rates throughout the summer. This is especially relevant at high elevation sites (Yala, Changri Nup). An interruption of the monsoon at 24K occurred in August 2016, possibly associated with an El Niño event (Kumar et al., 2006). During this interruption the energy balance returned to a pre-monsoonal regime of clearer skies, more pronounced diurnal temperature amplitudes, low precipitation rates and lower relative humidity (Figure A5), resulting in higher melt rates during that period (Figure 6e). Our analysis shows that some effects of monsoon are common for all surface types, while the presence or absence of debris and its thickness control how the incoming energy is absorbed and transmitted to the ice.

### 5.1.2 Debris-covered glaciers

Monsoon conditions in combination with debris cover affects the energy balance in a way that stabilizes melt rates over the ablation period, despite increasing air temperatures typical of monsoon (Figure 6). Enhanced melt during the night and morning in monsoon, resulting from higher $T_a$, is partly offset during the cooler afternoon hours (Figure A10) when the wet air masses usually arrive in this region, bringing intense cloud-cover and precipitation. While $H$ decreases as a consequence of a smaller average temperature gradient between surface and air, more latent energy is released from the wet debris surface, which shifts from a water-limited process during pre-monsoon to an energy-limited process during monsoon (Figure 8). It has been known from studies at individual sites that debris cover protects the ice by returning energy to the atmosphere in the form of turbulent fluxes (Yang et al., 2017) and that the turbulent fluxes can be a major component in the energy balance during both dry and wet conditions (Steiner et al., 2018). Accounting for the debris water content through the inclusion of a simple interception storage has allowed us to identify the importance of the latent heat flux in the cooling effect of evaporation from debris during monsoon, a process that has often been neglected in previous modelling studies. We have been able to quantify how cloud overcast and additional moisture modify the energy balance of debris-covered glaciers, and especially the turbulent fluxes, to result in a melt-equalizing effect.

### 5.1.3 Clean-ice glaciers

In contrast to debris-covered glaciers, when clean-ice glaciers are snow-free and the ice has been heated to the melting point, almost all net radiation leads directly to ice melt, while the turbulent flux contribution remains small. When entering the monsoon period, the latent heat flux switches sign, changing from sublimation to condensation, which adds energy to the surface instead of removing it. This behaviour has previously been observed at sites with a 'southern influence' (Yang et al.,





2017; Azam et al., 2014), but has not been indicated for the drier conditions on the Tibetan Plateau (Mölg et al., 2012; Sun et al., 2014).

### 5.1.4    Dirty-ice glacier

At our site with thin debris, the effects that we have observed for debris-covered and clean-ice sites combine to create a melt-enhancing effect that becomes particularly potent during monsoon: the dark debris surface absorbs almost $90\%$ of $SW_\downarrow$ in the case of Hailuogou. With a short conduction length, the energy influx leads almost directly to melt. The cold surface favours condensation and a strong temperature gradient between the surface and the air ($g_T$ pre-monsoon:$-2.08$, monsoon: $-2.61\,°C\,m^{-1}$). Driven by higher wind speeds (Table 6, Figures A9 and A7) and moist air during monsoon, both turbulent

fluxes ($H$ and $LE$) thus become potent sources of melt energy. This adds detail to prior observations and modelling results, that thin debris causes higher melt rates than at both clean-ice sites and sites with thicker debris cover (Reznichenko et al., 2010; Östrem, 1959; Reid and Brock, 2010; Fyffe et al., 2020), especially in humid environments (Evatt et al., 2015), e.g. the location of Hailuogou glacier.

### 5.2    Sensitivity of seasonal flux changes to elevation and debris thickness

Our results are derived from simulations at one location (AWS) on each glacier. To understand how representative they are of conditions across the glacier ablation zone at each site, and across the possible range of debris thicknesses in particular, we conduct a sensitivity experiment at each site. We re-run the model synthetically varying the AWS elevation to represent the range of elevation of each glacier ablation zone by applying a $T_a$ lapse rate of $0.6\,°C/100m$ and, for the debris-covered sites, by varying also the debris thickness in the range 10-80 cm (for ranges and steps see Table 7). Using the station-measured,

accumulated albedo is not appropriate during this experiment, due to changing snow conditions with varying elevation. We therefore include the parameterisation introduced by Ding et al. (2017) for modelling $\alpha$. From the resulting range of EB flux outputs, we calculate the range of expected changes for the entire ablation zone when moving from pre-monsoon to monsoon ($\Delta$-range). This allows us to place our results in the context of the changes that can be expected over the entire ablation zone, given its elevation span and debris thickness variability. Figure 9 shows that even accounting for the range of conditions across

each glacier ablation area, the pattern of pre-monsoon to monsoon difference in flux components, and importantly $dQ$, remain similar for debris-covered sites: The $\Delta$-range of $dQ$ stays within the uncertainty range, with the exception of Langtang, where the unrealistic combination of relatively thin debris and low elevation causes high $dQ$ $\Delta$-range. This lends confidence to the results obtained at the individual AWS locations. Although we adjusted forcing data for elevation in this exercise, we could not represent the effects of variable debris thicknesses in modifying $2m$ meteorological variables (Steiner and Pellicciotti,

2016; Shaw et al., 2016). This comes with the assumption that surface-atmosphere interactions are negligible compared to the altitudinal patterns and temporal changes. While this might be acceptable at thicker debris sites, it is more questionable at Hailuogou, where the observations were taken above thin and cold debris. However, also at this site, the $\Delta$-range ends up to be small ( $5\,W\,m^{-2}$) and close to zero when debris between 10 and $80\,cm$ thickness is applied artificially.



**Table 7.** Ranges of elevations and debris thicknesses used for the sensitivity runs, including the glacier terminus elevation (min), the AWS elevation (AWS) and the upper debris limit on debris-covered glaciers or to the approximated ELA elevation on clean-ice glaciers (max). We also show the range of debris thicknesses $h_m$ modelled for debris-covered glacier sites. All combinations of elevations and debris thicknesses were used.

| Glacier | | Lirung | Langtang | Yala | Changri Nup | 24K | Parlung No.4 | Hailuogou |
|---|---|---|---|---|---|---|---|---|
| **min** | $[m.asl]$ | 3990 | 4500 | 5170 | 5270 | 3910 | 4620 | 2980 |
| **AWS** | $[m.asl]$ | 4076 | 4557 | 5350 | 5471 | 3900 | 4800 | 3550 |
| **max** | $[m.asl]$ | 4400 | 5600 | 5400 | 5600 | 4200 | 5400 | 3700 |
| h_d $[cm]$ | 10, 20, 30, 40, 50, 60, 70, 80 | | | | | | | |

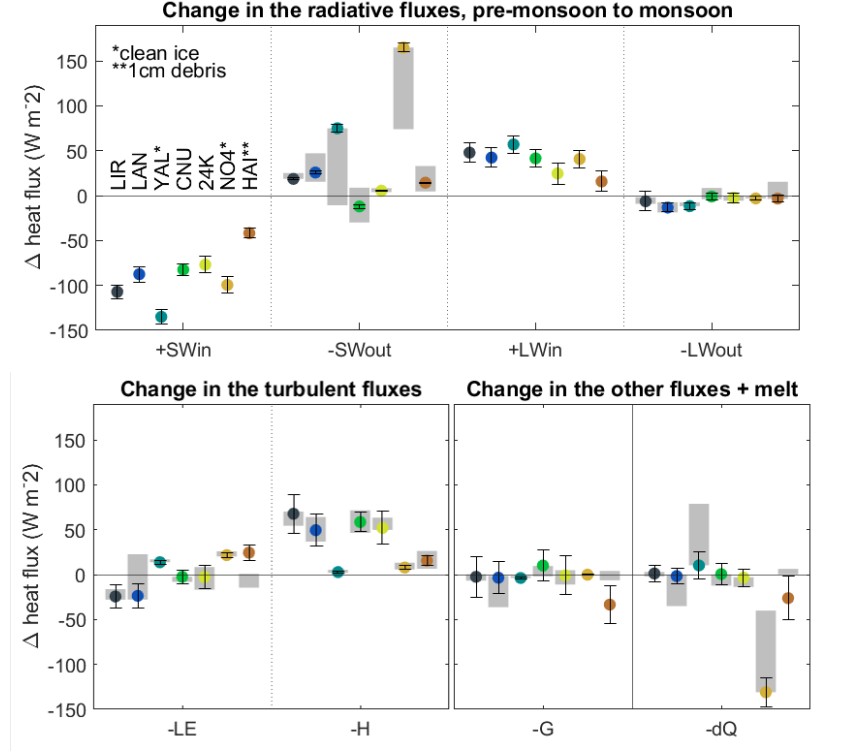

**Figure 9.** Changes in the individual fluxes when moving from premonsoon to monsoon. Color dots indicate 'standard' runs with AWS site specific conditions. Black bars indicate the uncertainty range on the standard runs. Grey indicates the sensitivity of flux changes (Δ-range) to debris thickness varied from 10 and $80\,cm$ combined with the elevation range of debris-covered ablation zone of the individual glaciers;

## 5.3 Limitations

Our modelling approach allowed us to constrain the energy and mass balance at the AWS location in the best possible way for the different glacier surface types. Two debris-specific parameters ($k_d$ and $z_0$) were optimised in two separate steps at the





debris sites (see section 3.3) and $\alpha$ was derived from measured $SW_\downarrow$ and $SW_\uparrow$ and given as an input at all sites. While this is not a feasible approach at the scale of entire glaciers, for application at other sites, or for future simulations, as it requires specific observations of radiation and ablation, both choices were made here in order to constrain the energy balance with

observations to the greatest degree possible. Requisite data (e.g. thermistor strings, wind towers, high-resolution topography) to estimate these two parameters with established methods were not uniformly available across our study sites. These observational methods themselves can be sensitive to the measurement period (e.g. Nicholson and Benn, 2006) and are subject of ongoing research (e.g. Rounce et al., 2015; Miles et al., 2017; Rowan et al., 2020; Quincey et al., 2017), so we concluded that the calibration approach here was best suited to the application at all sites. In this respect, all optimized values fall within the

expected range based on prior energy-balance studies on debris-covered glaciers (Yang et al., 2017; Rounce et al., 2015; Evatt et al., 2015; Reid and Brock, 2010; Nicholson and Benn, 2006; Rowan et al., 2020; Lejeune et al., 2013; Collier et al., 2015; McCarthy, 2018), but consistent measurements of debris parameters remain a community research priority to ensure robust modelling of sub-debris melt. To represent evaporation and the latent heat flux from the debris, we assigned an interception storage of $2\,mm$ water to the debris surface, as, to the authors' knowledge, very few measurements have been collected of the

debris layer's interception capacity and moisture content. This is important because the thermal properties of debris vary with moisture fluctuations inside the debris (Nicholson and Benn, 2006; Collier et al., 2014; Steiner et al., 2018), so the moisture retention properties of the surface debris layer may play an important role in regulating energy transfer. However, these thermal properties have been found to be stable during the core monsoon months (Rowan et al., 2020), so we kept $k_d$ constant in time at each site in our model. Additional, targeted investigations should examine the possible role of water storage and mobility

within debris layers (Giese et al., 2020; Collier et al., 2014).

### 5.4 Implications

Our results show that the surface type plays a large role in modulating a glaciers' response in the seasonal transition from premonsoon to monsoon: we find a melt-enhancing effect of thin debris, driven by enhanced energy uptake attributable to low albedo and turbulent energy fluxes. We also find that, on clean-ice glaciers, $LE$ switches seasonally to become a source instead

of a sink of energy to the surface, while net radiation determines melt. Importantly, we additionally identify a melt-equalizing effect of debris cover under monsoon conditions at a number of sites. Monsoon-influenced, summer-accumulation glaciers (such as Langtang, Lirung, Yala, and Changri Nup) have been previously shown to be especially vulnerable to warming due to a decrease in accumulation and an enhancement of ablation with lowering albedo (Fujita and Ageta, 2000), and our results confirm that net shortwave radiation is the key control on monsoon-period melt rates for clean ice glaciers. Our results also

emphasize that the longevity of pre-monsoon snowcover into the monsoon period is a key control on melt rates, supporting past findings that the strength and timing of the monsoon onset has a profound impact on small mountain glaciers (Mölg et al., 2014, 2012) through the phase change of precipitation in the transition to monsoon conditions (Fujita and Ageta, 2000; Ding et al., 2017; Zhu et al., 2018).

The distinct responses of the surface energy balance of debris-covered and clean-ice glaciers to the pre-monsoon to monsoon

transition potentially accumulate to large mass balance biases when not adequately taken into account in models. It is thus im-





portant to also consider how this seasonal transition may evolve under changing climate, and what the consequences for glacier mass balance might be. All future model simulations agree on continued warming during the 21st century over High Mountain Asia, together with a strengthening of elevation dependent warming (Palazzi et al., 2017) and increases in moisture availability. An analysis on the ensemble estimates of regionally downscaled CMIP5 projections (CORDEX) for the Himalayas (Sanjay

et al., 2017) shows that total summer precipitation is projected to increase for 2036-2065 (2066-2095) by 4.4% (10.5%) in the Central Himalaya and by 6.8% (10.4%) in the Eastern Himalaya under RCP4.5 scenarios, relative to 1976-2005. While there is broad model consensus on the increase in future precipitation, there is little consensus on the future variability, frequency and spatial distribution of precipitation across High Mountain Asia (Kadel et al., 2018; Sanjay et al., 2017), which is likely a result of complex and poorly understood drivers of past monsoon changes (Saha et al., 2014; Saha and Ghosh, 2019), coarse

resolution of the baseline products and topographic variability in the region (Sanjay et al., 2017). A slight shift towards an earlier monsoon onset of <5 days over the coming century together with an increasing shift towards a later retreat by 5 to 10 days (mid-century) and 10 to 15 days (end-century) might increase the length of the monsoon period, with stronger lengthening in the Eastern Himalaya (Moon and Ha, 2020).

The prospect of warmer temperatures together with increased precipitation would (1) cause a shift in the precipitation partition

from snow to rain in the monsoon, resulting in snow cover shifting to higher elevations and increasing total melt; (2) potentially lead to an increase in early spring snowfall, which would delay the onset of ice melt; (3) increase the likelihood of ephemeral monsoonal snow cover but move it to higher elevations, thus leaving more of the lower ablation zones exposed; (4) increase the wetbulb temperature together with humidity to result in a reduction of the solid fraction of precipitation during monsoon. Overall it is likely that glacier ablation zones will be exposed for longer periods under future climate due to a net decrease of

the snow covered duration, with a resulting increase in total ablation. A lengthening of the monsoon into autumn, on the other hand, (Moon and Ha, 2020) would somewhat offset warmer air temperatures with regards to the late-season melt for all glacier types.

Warmer and wetter monsoonal conditions, including increased cloudiness, are likely to result in an overall increase of melt rates on clean-ice and dirty-ice glaciers. This is because (1) they are more directly controlled by $R_{net}$, which is likely to increase

in magnitude; (2) the turbulent fluxes towards cold surfaces are also likely to increase in magnitude, and they tend to 'work against' these types of glaciers. In contrast, the turbulent fluxes 'work for' debris-covered glaciers, and the melt-equalizing effect of debris under monsoon would likely remain in place. With these components summing up to have an overall protective effect on debris-covered glaciers, they are likely to resist the projected changes in the monsoonal summer longer into the future. Previous studies suggested that the mass balance of DCGs might be less sensitive to climate warming than clean-ice glaciers

(e.g. Anderson and Mackintosh, 2012; Wijngaard et al., 2019; Mattson, 2000). Here we confirm this hypothesis and suggest that this is even more the case under monsoonal conditions.

Simplified methods of glacier melt calculation (e.g. relying only on temperature or temperature and shortwave radiation) may integrate some of these processes through calibration (e.g. Ragettli et al., 2016). However, in a study on two clean-ice glaciers (Yala and Mera), Litt et al. (2019) were able to transfer Temperature Index and Enhanced Temperature Index models between

sites and years only during pre-monsoon. During monsoon, the transfer failed due to site-specific and inter-annually variable





cloudiness.

## 5.5 Future work

By modelling seven different study glaciers across the Himalayan arc, we have been able to identify important monsoon effects
on the energy and mass balance of debris-covered and clean-ice glaciers. In the context of the large spatial and inter-annual
variability in the climate, including the monsoon strength and timing, our sample size remains small due to data availability.
The timing and quantity of spring snow-cover and monsoonal snowfalls have large effects on the energy and mass balance,
and these variables can be highly variable between years. A future analysis could leverage a greater number of complete AWS
records, and possibly multi-year records, in order to extend some of our findings. Despite the advanced representation in Tethys-
Chloris, the surfaces of glaciers, and especially of debris-covered glaciers, are more complex systems than indicated at the AWS
location. Future work should be invested into spatially distributing forcing data and parameters necessary to run energy-balance
models in a distributed framework at glacier and catchment scales. Studies employing distributed energy balance models could,
for example, test the melt-equalizing effect of debris at the glacier scale and its overall effect on the catchment runoff. This
could be combined with generating spatially consistent forcing data using high-resolution weather modelling as in Bonekamp
et al. (2019) or Potter et al. (2020), and expanded to larger domains. Realistic representations of the glacier surface, including
distributed debris thickness, and supraglacial features, such as ice cliffs and surface ponds, should also be established to provide
well-constrained water budgets and improved representations of glacierised environments in land-surface models under present
and future environmental conditions.

## 6 Conclusions

By modelling the energy and mass balance of seven glaciers in the Himalayas at on-glacier weather stations, we identify and
explain the main effects on the energy and mass balance caused by the Indian and East Asian summer monsoons. Heavy cloud
cover, liquid precipitation, wind speed, the presence and thickness of debris cover, and elevation shape the energy and mass
balances of the Himalayan glaciers during the ablation season. The timing of snow melt-out and the presence of ephemeral
monsoonal snowcover play a particularly important role, especially at high elevations. We highlight key pre-monsoon to mon-
soon changes in energy fluxes, distinct for clean-ice and debris-covered glacier sites: the melt of clean-ice glaciers is primarily
radiation-driven at any point during the ablation season, and strongly influenced by albedo variations. The latent heat flux can
initially be a small sink and turn into a source during the core monsoon. Debris cover can act in two ways: (1) Once a debris
layer of a few centimetres is established, it returns most of the heat it absorbs back to the atmosphere via longwave and turbulent
heat fluxes. The sensible heat flux reduces during core monsoon, but the latent heat flux removes energy from the surface due
to evaporation of liquid water. The turbulent fluxes readjust in this way to monsoonal temperature, wind and moisture regimes,
maintaining a nearly constant melt rate over the entire ablation period. Sensitivity analyses of our energy-balance model shows
that these findings hold for large portions of the debris-covered glacier surfaces. (2) When it is thin, debris amplifies the in-





coming energy due to its dark and cold surface, small conduction distance to the ice and sensitivity to turbulence. The turbulent fluxes 'work for' the debris-covered glaciers by returning absorbed energy to the atmosphere, and tend to 'work against' clean-
ice and dirty-ice glaciers under monsoonal conditions. We thus expect the mass balance of debris-covered glaciers to react less sensitively to projected future monsoon changes than clean-ice and dirty-ice glaciers.





*Code and data availability.* Model and analysis codes as well as AWS datasets are available upon request during the discussion and review phase and will be made publicly available at a later stage.

*Author contributions.* SFu, FP and EM designed the study. SFu carried out the analysis with the help of CF, MM and SFa. SFu interpreted the results, created the figures and wrote the manuscript with the help of CF, EM, MM, TS and FP. SFa, PW, WI, and QL reviewed the manuscript. WY and BD facilitated field data collection and provided parameterisations for albedo and precipitation phase. WY, PW and WI also contributed data sets.

*Competing interests.* The authors declare that they have no conflict of interest.

*Acknowledgements.* This project has received funding from the European Research Council (ERC) under the European Union's Horizon 2020 research and innovation programme (grant agreement No 772751), RAVEN, "Rapid mass losses of debris covered glaciers in High Mountain Asia". We would like to thank Jakob Steiner and ICIMOD for hosting and contributing data sets. We also thank Marin Kneib for organizing and helping with field data collection and Achille Jouberton for commenting on the manuscript.





## Appendix A

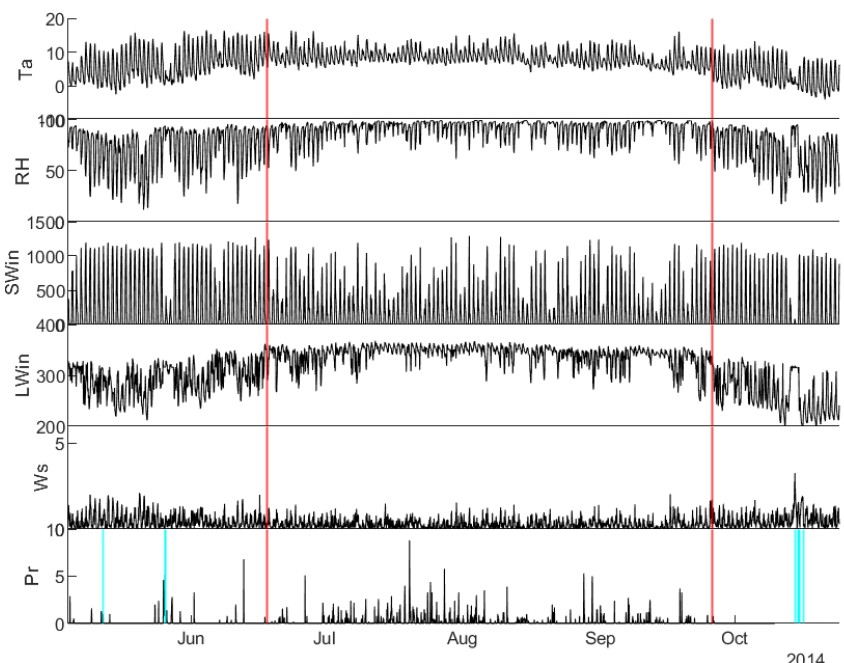

**Figure A1.** Meteoroligical observations on Lirung during the ablation season recorded by AWS; Red vertical lines indicate monsoon onset and end; Blue indicates time steps with snow cover at the AWS location, as determined from $\alpha$ (>0.5)

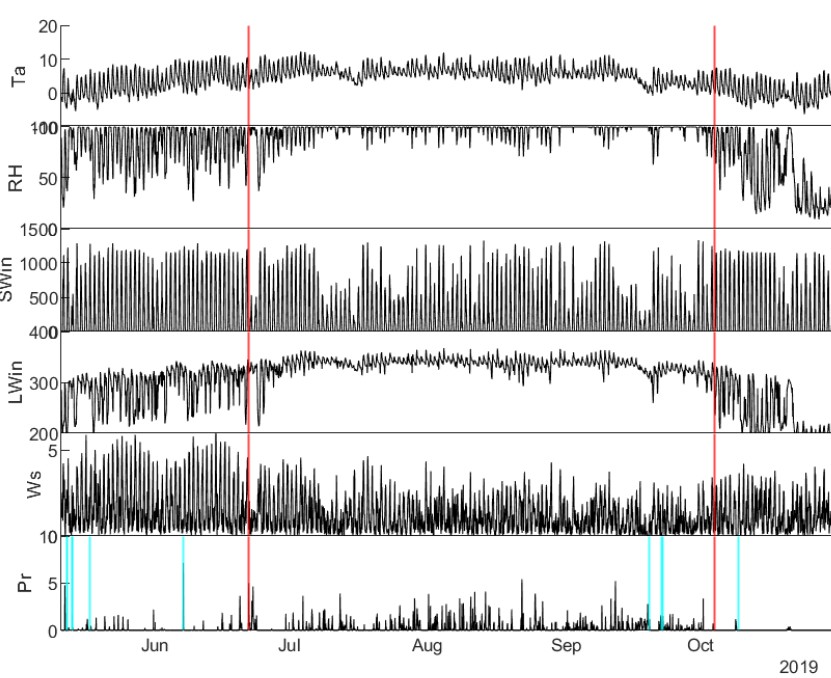

**Figure A2.** Meteoroligical observations on Langtang during the ablation season recorded by AWS; Red vertical lines indicate monsoon onset and end; Blue indicates time steps with snow cover at the AWS location, as determined from $\alpha$ (>0.5)




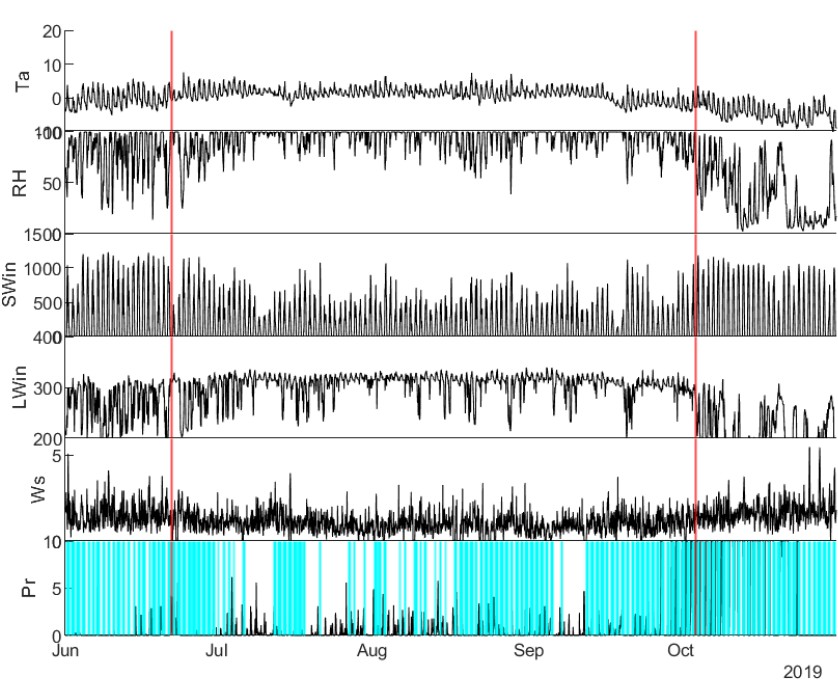

**Figure A3.** Meteoroligical observations on Yala during the ablation season recorded by AWS; Red vertical lines indicate monsoon onset and end; Blue indicates time steps with snow cover at the AWS location, as determined from $\alpha$ (>0.5)





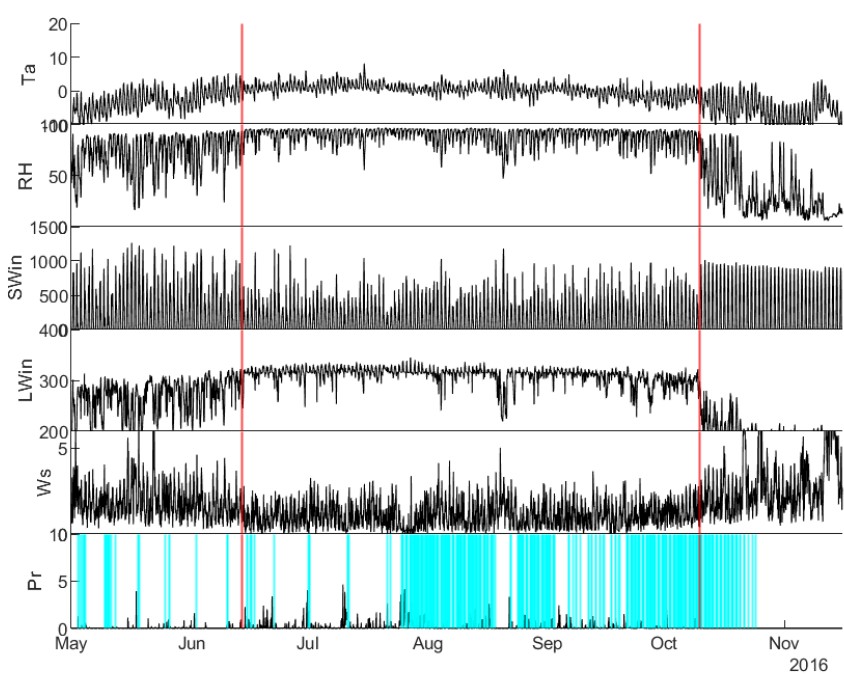

**Figure A4.** Meteoroligical observations on Changri Nup during the ablation season recorded by AWS; Red vertical lines indicate monsoon onset and end; Blue indicates time steps with snow cover at the AWS location, as determined from $\alpha$ (>0.5)





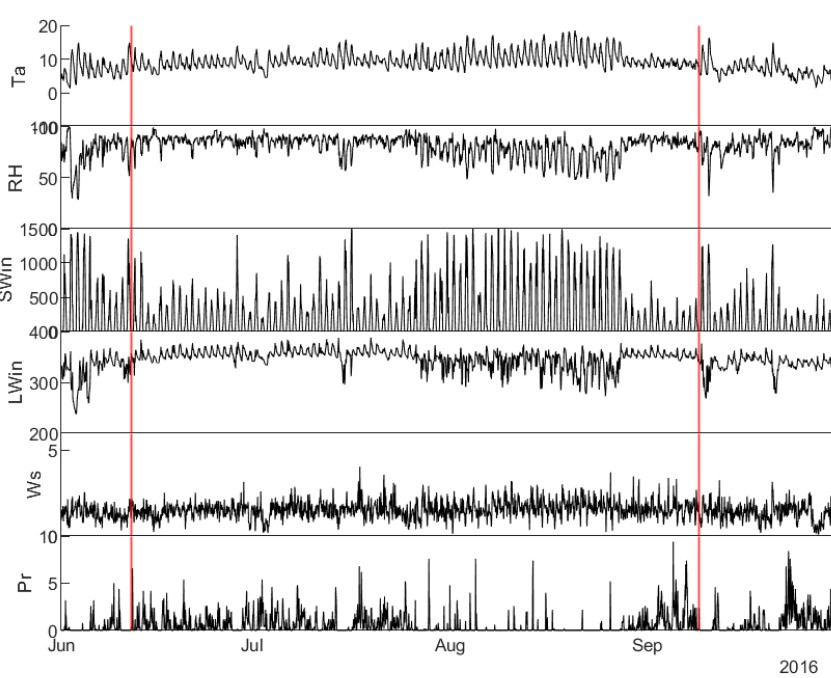

**Figure A5.** Meteoroligical observations on 24K during the ablation season recorded by AWS; Red vertical lines indicate monsoon onset and end; Blue indicates time steps with snow cover at the AWS location, as determined from $\alpha$ (>0.5)





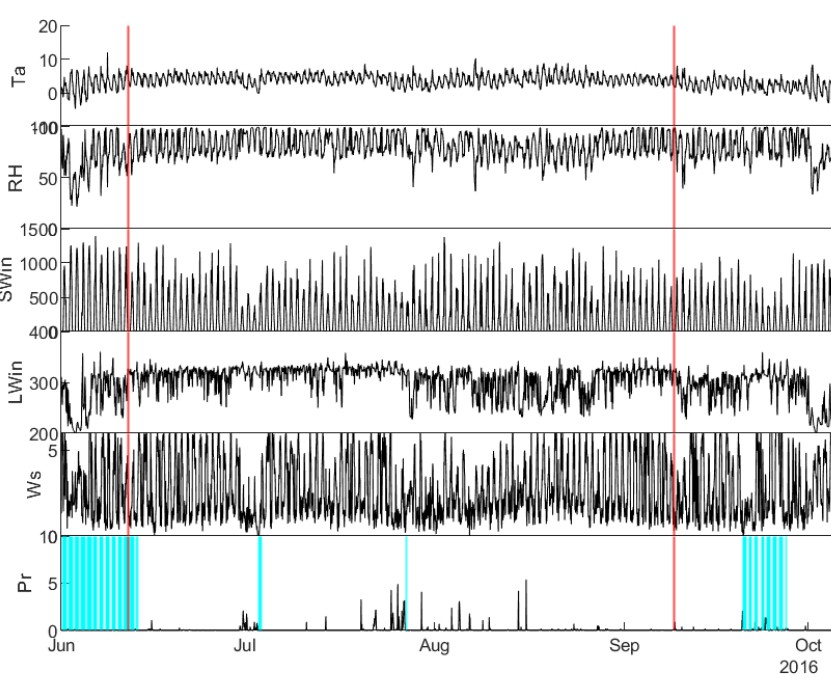

**Figure A6.** Meteoroligical observations on Parlung No.4 during the ablation season recorded by AWS; Red vertical lines indicate monsoon onset and end; Blue indicates time steps with snow cover at the AWS location, as determined from $\alpha$ (>0.5)

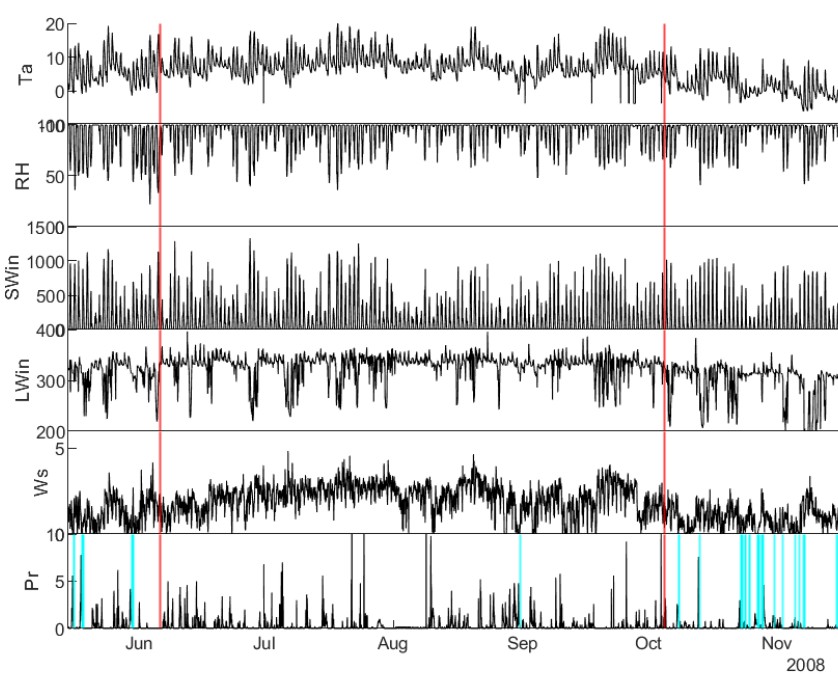

**Figure A7.** Meteoroligical observations on Hailuogou during the ablation season recorded by AWS; Red vertical lines indicate monsoon onset and end; Blue indicates time steps with snow cover at the AWS location, as determined from $\alpha$ (>0.5)


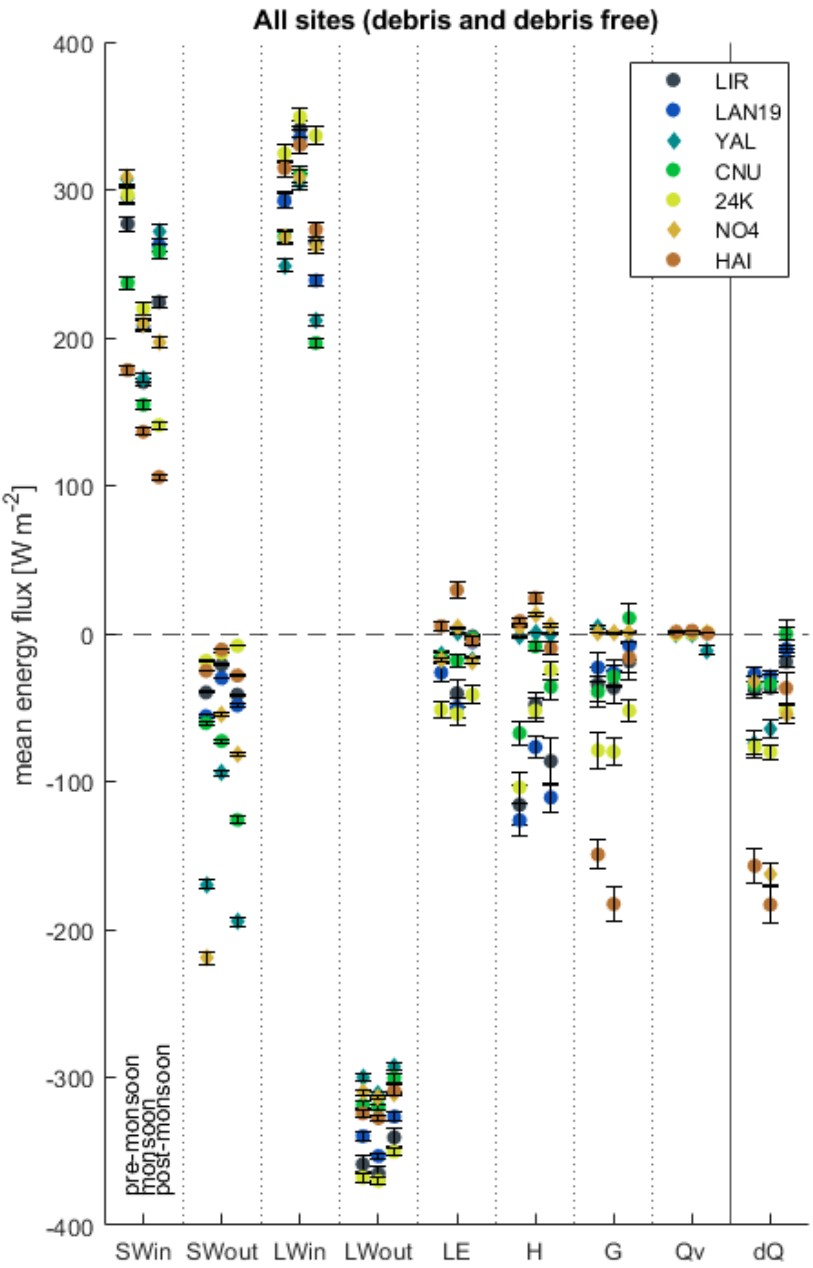

**Figure A8.** All energy balance components of all glaciers in comparison, split into pre-monsoon, monsoon, post-monsoon; black bars indicate the uncertainty range;



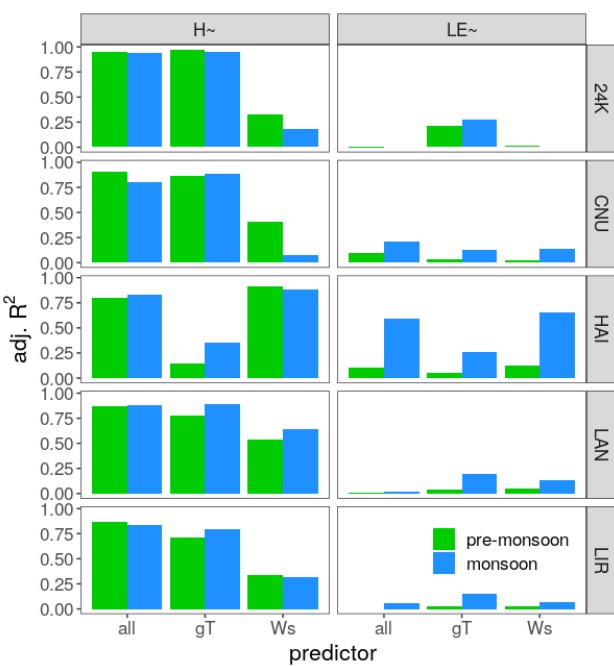

**Figure A9.** Predictive power of temperature gradient between surface and air ($gT$) and wind speed ($Ws$) and their combination for determining $H$ and $LE$. 'All' indicates a multiple polynomial regression model using both variables, otherwise a univariate polynomial regression model was used, and both models had 2 degrees of freedom.

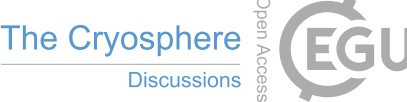

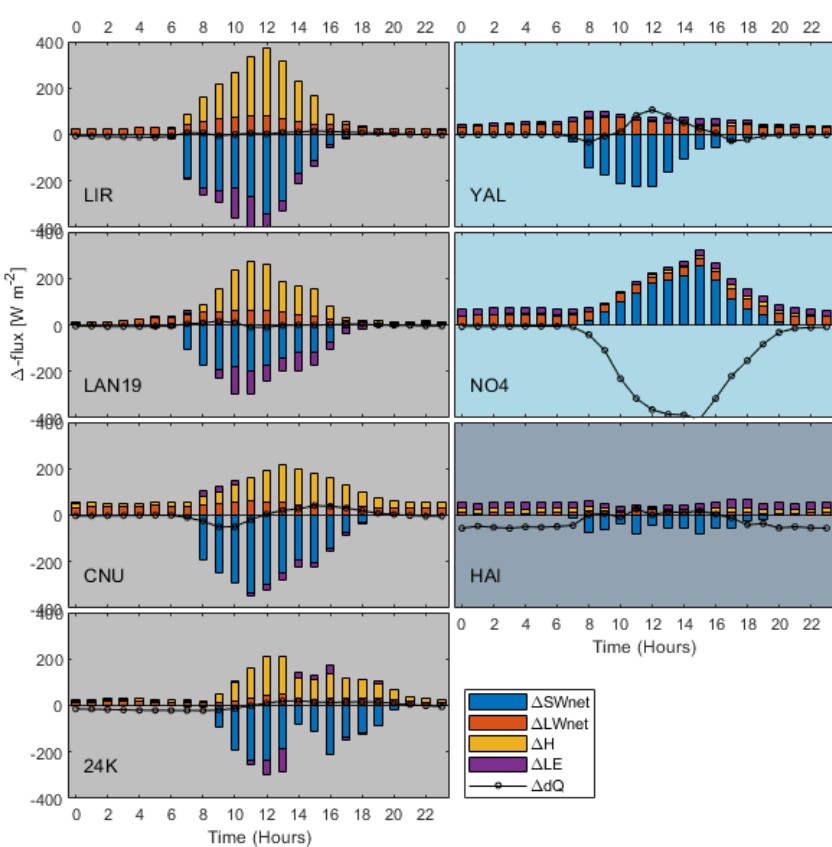

**Figure A10.** Energy flux differences in the diurnal cycle (stacked) between pre-monsoon and monsoon; The direction of change is to be considered relative to the sign of the orginial flux. Fluxes towards/away from the surface have a positive/negative sign. For example, a positive change in a negative flux means a reduction in the magnitude flux, and can also lead to a change in sign. Grey background indicates debris covered site; light blue indicates clean ice sites; grey-blue indicates 1cm debris site



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
