# Peer review of "Understanding monsoon controls on the energy and mass balance of glaciers in the Central and Eastern Himalaya"

_The Cryosphere, 2021_

## Author Comment (AC1)

**Answers to Referee #1**

We thank the referee for their useful comments. We answer all comments point-by-point below each statement in blue font.

Review on the manuscript entitle "Understanding monsoon controls on the energy and mass balance of Himalayan glaciers'

General comments:

Overall manuscript has provided a comprehensive study of the glacier energy and mass balance for seven sites and further generalized for the whole Himalaya. That may be the reason; the title has come up with Himalayan glaciers. However, this study has focused on the only on the seven glaciers and circled around the Nepal Himalaya and Tibetan Himalaya. So Eastern Himalaya is more appropriate.

We thank the reviewer for acknowledging the comprehensive nature of our study. Our intention was not to generalize our results for the whole Himalaya and we are sorry if the title suggests this. We agree with the reviewer and will change the title to "Understanding monsoon controls on the energy and mass balance of glaciers in the Central and Eastern Himalaya". We think however that it is appropriate to use both Central and Eastern Himalayas, following Bolch et.al (2012), where "Central Himalaya"encompasses mostly the Nepalese Himalayas, and where "Eastern Himalaya" and "Western Himalaya" refer to the regions to the east and west of the Nepalese Himalayas, respectively. By moving "glaciers" in front of "Central and Eastern Himalaya", we hope to reduce the impression of generalization additionally.

There is several new information which is really valuable for the understating the summer accumulation type glaciers. One of that is: At all sites, ice melt is the dominant mass loss component, accounting for 65.8% (Changri Nup) to 95.4% (Hailuogou,) of the total mass losses.

We thank the reviewer also for appreciating the novelty of our results. Regarding the numbers that the reviewer cites and from which it is evident that ice melt is the dominant mass loss component: this is so as we have only considered the mass losses during the ablation period for which measurements are available (May to October for Changri Nup and mid-May to October for Hailuogou). The numbers for the year-round mass losses would include a greater snowmelt component. We will modify the Results section of the manuscript in the way shown below, to make sure this is clearer to the reader:

The ablation season average melt rates vary considerably across sites, with the highest value of $42.7\,mm\,d^{-1}$ reached at the low-lying thinly debris covered site, Hailuogou, and the lowest value of $6.1\,mm\,d^{-1}$ reached at Langtang, a site at moderate elevation but the thickest debris cover (Figure 4). The largest average seasonal mass loss component at all sites is ice melt, with a minimum of 65.8% of the mass losses at Changri Nup (Figure 4c), up to 95.4% at Hailuogou, (Figure 4g). This is followed by snowmelt, accounting for only 0.1% at 24K (Figure 4e) but as much as 33.1% at Yala (Figure 4c) of the seasonal mass losses. Sublimation from ice and snow represents a very small share of the seasonal mass losses, and ranges from 0.01%

...

Few more general comments, in fact it is query to be generalize.

The manuscript only talks about pre-monsoon and monsoon period, what about post-monsoon? Does it differ from pre-monsoon?

We have also analysed the post-monsoon period and compared it to the other two seasons. The main reason why we did not include this analysis in the manuscript is that the AWS data were unreliable at the two highest sites for the post-monsoon (Yala after mid-September and Changri Nup after August), especially with respect to the precipitation and snow depth measurements. We were concerned that this would bias the resulting energy and mass balances. Although we had multi-year timeseries at hand, the chosen years for those two sites contained the most complete records of an ablation season. We have described this for Changri Nup in section *4.1 Modelled mass balance* and will also add this description for Yala. We also felt that the paper already contains extensive results that allow identifying the distinct characteristics of the monsoon. The paper is already dense and contains much information (see e.g. Referee #2 comments)  and both figures and text would become overly complex if we also added the post-monsoon (this necessitates two additional comparisons, pre/post and monsoon/post). We decided to refrain from including an analysis on the post-monsoon into the main text, and hope that the reviewer will agree with us.

(ii) There is no discussion about the effect of winter precipitation on the energy and mass balance of the glaciers. Although the manuscript deals with understanding monsoon controls on energy balance and mass balance, but winter precipitation has equal control over the energy and mass balance.

We fully agree with the concerns the referee raises: the timing and quantity of winter and spring snowfalls greatly shapes the **annual** energy and mass balance through the albedo effect. However, while an analysis of the influence of winter precipitation would be a worthwhile analysis, it goes beyond the scope of the present study, which focuses on identifying the influence of monsoonal conditions on the ablation season glacier energy balance.

Section wise comments:

L2: "large temperature amplitudes" make it simpler like large temperature ranges.

This is a good suggestion and we will modify the text.

L5-6: This sentence, I would like to see at the end of the introduction, where citation of work may validate it.

We agree that this might not be an appropriate sentence for an abstract. As the last paragraph of the introduction started with a sentence of similar content, we will remove it from the abstract, and instead stress the importance of energy balance studies in a shorter sentence. *"Glacier energy and mass balance modelling using in-situ measurements can offer insights into the ways in which surface processes are shaped by climatic regimes"*.

L7: 'Himalayas' it is for curiosity on using 'The Himalayas' instead 'The Himalaya'. I am actually not sure which one is better.

We shared the reviewer perplexity here. Looking this up, according to the word origin (Sanskrit, "hima" = snow, "alaya" = abode), the name refers to the mountain range as an

"abode of snow". Thus, from the etymological perspective, the singular "Himalaya" is more appropriate. In modern times, it was misinterpreted as referring to the single mountain, hence all the Himalayan mountains together were turned into the plural "Himalayas". In published cryosphere literature, both writings are frequently used, so we decided to use the etymologically more correct way, and will switch to "Himalaya" throughout the manuscript. We thank the reviewer for this hint!

L 19: "dirty-ice glaciers", somewhere it was mentioned as thin debris, so does the dirty-ice glaciers are the same ?. If so then thin debris is mostly lies over the patches or around the higher elevation. Whereas, it has mentioned here as dirty-ice glaciers, which what I understand is that the whole glacier has dirty-ice only.

We revisited these definitions and realized that "dirty ice" may after all not be the right term to describe the surface of Hailuogou glacier's ablation zone. According to Fyffe et al. (2020) dirty ice is only "partially debris-covered", "patchy" or "discontinuous". According to our own field-observations, Hailuogou's ablation zone is to a large extent continuously covered with a thin layer of fine clasts and scattered with coarser clasts, which would leave the thin layer visible, and directly influenced by the atmosphere. Co-author Liu Qiao, who has maintained an AWS on Hailuogou between 2008 and 2013, has measured a debris thickness of 1cm at the AWS site. We have therefore decided that using the definition "thin debris" is more appropriate and will remove the use of "dirty ice" everywhere. We will also revise the description of Hailuogou glacier in L120-121 and remove the mention of dirty ice in the *Introduction* section and a related citation in L61.

L 21: (Yang et al., 2017), please check.

We will remove this. This is an artefact in our LaTeX code.

L28: "Karakoram, Pamir and Kunlun ranges in the east". I think it should be 'west'.

We will correct this mistake.

L55-57: This has no information except to show that these researchers have published work on debris-covered glaciers.

We respectfully disagree here, because this citation lists all the studies introducing energy balance models for debris-covered glaciers and thus represents the evolution and state of the art of this type of model.

L62-63: In continuous to the pervious comments. Here are some other references having in situ observations on the central Himalayan glaciers with the perspective of debris cover and thickness influences on ice melt (Shah et al., 2019 and Pratap et al., 2015).

Thank you for these suggestions. We already cited Shah et al. (2019) in L60, who conclude that debris thickness has a stronger control on glacier melt than elevation. We however missed Pratap et al. (2015) and will now also include this study.

L73-75: this whole paragraph, I dint see any sense before to define the objectives of this study.

We agree that this part seems disconnected and interrupts the flow of the introduction. It does also not contain essential information for motivating the analysis, so we decided to remove it from the text.

L87: 'glacierised' I generally practice to use 'glacierized' as per Cogley et al., 2011 (glossary of glacier mass balance and related terms).

Both the US American ("glacierized") and British ("glacierised") spellings are accepted and used in the literature. We decided to generally adopt English spelling in the manuscript and thus will keep "glacierised".

L92: Table 2 cited before Table 1, check it with journal style.

Thank you for spotting this. We will change the order of these two tables.

L104: This might be the ablation area that has disconnected from the accumulation area. if this is the case then in the Table , the Lirung Glacier's characteristics needs to be revised.

We are not sure we understand the reviewer's comment here and would kindly ask him/her to clarify how we should revise Lirung glacier's characteristics in Table 1. Currently, the table contains the characteristics of both the accumulation area and the (dynamically disconnected) ablation area together, e.g. the sum of the areas of both glacier parts. We will make this clear in the caption.

Figure 2: Caption: "(blue bars)" For me the color is aqua and not blue. "area on the x-axis [km2] and altitude on the y-axis [m.asl]", This information isn't shown in the figure. Area (size) of the glaciers is not clear, therefore additions of a scale bar and direction arrow is required. "Black crosses" this sign need to change as at Yala Glacier it entirely covers the glacier. Make it red dot with AWS on the side as a legend.

We agree that "aqua" is a more suitable name of the color. We also changed the color name for the area/bars indicating debris cover to "olive". The glacier area is expressed in 100m elevation bands in the diagram. We agree that it is not easy to judge the glacier size and orientation without a scale bar and direction arrow. We now added these elements to the figure. We also decreased the size of the x-indicators and arrows for better readability and added a legend, but we kept the black color for contrast and style reasons.

[Figure]

*Figure 2 revised*

L134: The figure description is not in order.

We will change the sequence of the text, so that the references (a-e) are called in order. Based on comments from Referee #2, we will move this part "*Climatic and meteorological conditions*", including Figure 3, to the supplementary.

L138: 1st if one consider the lirung and yala glaciers with an elevation difference 1000 m asl in the same catchment, and 2nd by including the fully debris covered ablation area and other clean ice , how it can be justify that the mean monthly 2 m Ta is very similar on the both sites. Though, it is an observation (Fig. 3a) but just to rethink.

We thank the referee for this useful and insightful comment. The referee is very right that the similarity in air temperatures between Lirung and Yala glaciers is unrealistic for precisely the reasons the referee mentions. But please consider that the climatology for each glacier in Figure 3 is taken from one gridcell of 9x9km horizontal resolution of the ERA5-Land reanalysis product and represents average conditions within this gridcell, and not the conditions at the actual elevation of each glacier. We refrained from adjusting the ERA5-Land outputs, as the purpose of this figure is only to show that the study year for each site falls well within the typical interannual range – these data would need careful downscaling to adequately represent on-glacier conditions. We will acknowledge this circumstance in L130-133: "*Here, we use the monthly averaged ERA5-Land reanalysis data (Muñoz Sabater, 2019) to provide an overview of the long term climatic patterns, ... and evaluate the representativeness of the AWS records in terms of seasonal variability ..., while acknowledging that the absolute values from the reanalysis dataset might be biased.*"

*Motivated by your comment, we will add to this sentence "... and do not represent the conditions at the glacier location"*

L192: "surface temperature Ts". Please elaborate that how Ts was calculated?.

*The reviewer is right that our formulation was confusing. The calculation of surface temperatures was explained in L176 - 179 "To close the energy balance, a prognostic temperature for the different surface types ($T_{sno}$, $T_{deb}$, $T_{ice}$) is estimated for each computational element. Iterative numerical methods are used to solve the non-linear energy budget equation until convergence for the ice and snow surface, and the heat diffusion equation for the debris surface, while concurrently computing the mass fluxes resulting from snow and ice melt and sublimation."*

*We did not inform the reader however that $T_{sno}$, $T_{deb}$, $T_{ice}$ are equivalent to $T_s$ in the equations that are not specific to a surface type.*

*We will add a sentence clarifying the use of the symbols:*

*"In the case of snow, debris and ice surfaces, $T_{sno}$, $T_{deb}$ or $T_{ice}$ are equivalent to the element's overall surface temperature $T_s$. In the following, we use the surface type specific symbol for surface specific equations, while we use $T_s$ for equations valid for all three surface types."*

3.2 Mass balance in T&C. if it is the same name used before, i would suggest to use T&C model throughout.

*We will change to use "the T&C model" everywhere.*

L312: delete 'We vary'

*Unfortunately, we do not understand why "We vary" would be unnecessary here. As there might be a confusion around the word "vary", we will use the word "perturb" instead.*

L340: choose other word as it was already used with Tibetan plateau.

*We tried to find an alternative, but found no other word that would describe our observation as precisely. "plateau" is used as a verb here, while in "Tibetan Plateau" it is used as a noun or name.*

Figure 4. Caption and legend.

Measured and Obs., change to single. Black circles seems to be black dot.

*Thank you for pointing us at these inconsistencies. We will make the changes in the caption and legend.*

Figure 5. (i) what is the reason for using different color scheme for same component. I cannot differentiate the ice melt and sublimation for the LIR glaciers. I think use of single color like for LAN glacier would be ok.

*This is a good question. We gave each study site its own color signature throughout the manuscript. We were hoping that this would help the reader to intuitively recognize the study site by color in addition to the name. We had the experience in earlier studies, that using only the name of several study sites would sometimes confuse the reader, and the reader would have to spend extra time to repeatedly relate the name to e.g. the geographic location.*

We will change the color indicating sublimation to allow for an easier differentiation between sublimation and ice melt.

[Figure]

**Figure 5** *revised*

L481: "applying a Ta lapse rate of 0.6°C/100m" What about the change of values of other forcing variables with the change in elevation?

This is a very good comment. We considered all possible options for the extrapolations of the meteorological variables. While temperature has a relatively stable elevation lapse rate, which has been investigated and quantified in a number of studies, the other variables are not simple to extrapolate across the glacierised area (as for precipitation or wind speed), or the change over the glacier area was expected to be small (as for incoming shortwave radiation). For the purpose of this sensitivity exercise, we assumed that the strongest changes in meteorological forcing with elevation would be the air temperature, which in turns controls the precipitation partition and the albedo. To reduce the content and complexity of the main manuscript, and in response to comments from Referee#2, we will move the *Section 5.2 Sensitivity of seasonal flux changes to elevation and debris thickness* to the supplementary. We will, however, make this justification clearer in the supplementary. We note that this experiment does not affect any of the main paper results, which all derive from simulations forced with unadjusted AWSs data. The experiment goal was to ascertain that the results did not depend on the specific elevation and debris thickness of our AWSs.

L585-89: More things are also to be considered for realistic simulation, for example avalanches, crevasse, blowing snow, water channel, etc.

Yes, we agree with the need for these additional aspects of complexity, many of which are possible in the distributed implementation of T&C. We will remove this sentence in order to cut down on content. In L579, we already noted that *"the surfaces of glaciers, and especially of debris-covered glaciers, are more complex systems than indicated at the AWS location"*, which seems to be sufficient to make this point.

Bolch, T., Kulkarni, A., Kääb, A., Huggel, C., Paul, F., Cogley, J. G., ... & Stoffel, M. (2012). The state and fate of Himalayan glaciers. *Science*, *336*(6079), 310-314.

https://doi.org/10.1126/science.1215828

---

## Author Comment (AC2)

**Answers to Referee #2** on the manuscript *Understanding monsoon controls on the energy and mass balance of Himalayan glaciers.*

We answer all comments point-by-point below each statement in blue font.

This is an interesting concept and topic of research and a number of new analyses are presented in this manuscript. But I find that I am overwhelmed by the lack of synthesis in the analysis and the writing.
It is difficult to tie the in situ weather station and modeling results with the actual conclusions stated. This is partly because the paragraphs seem to jump from one flux to another or from one variable to another from sentence to sentence. I am left wondering if these conclusions are actual supported by the work in the manuscript or are rather just assertions? They might be but the figures, analysis and writing do not clearly support the conclusions in the discussion/ conclusions section.

We thank the referee for their comments. To address the referee's main concern regarding a lack of synthesis, we will introduce a number of improvements to both text and figures to explicitly link our results to the interpretations and conclusions. To provide the manuscript a logic thread and focus, we will formulate up front (in the revised *Introduction*) a clear set of research questions, replacing the more vague "research objectives" of the submitted version.

The new research questions are
1) Which energy and mass fluxes dominate the seasonal mass balance of glaciers in the Central and Eastern Himalaya?
2) How does debris modulate the ablation season energy balance in comparison to clean-ice surfaces?
3) How does the monsoon change the glacier surface energy balance?

We will give the manuscript a new structure and organise both results and discussion around the research questions listed above. The *Discussion*, in particular, will be structured to respond to each of them separately. The new overall structure of the manuscript will be as follows and is explained in details below:

Results:

First, in order to make our *Results* more structured and to link them to the revised figures systematically, we will adjust them as follows:

- We will maintain the subsections *Modelled mass balance* and *Modelled energy balance*. We will shorten these sections to focus them only on results required to answer our research questions, e.g. common energy balance patterns for all sites and the role of snow accumulation
- We will move the model evaluation from the *Results* (it was described originally in the *Modelled mass balance* section) to the *Methods*.
- we will introduce two new subsections: *Impact of debris cover* and *Impact of the monsoon,* to separate those aspects, and we will split the latter into three subsections, one each for surface type: *Impact of the monsoon on clean-ice sites, Impact of the monsoon on debris covered sites* and *Impact of the monsoon on thinly debris covered sites.*
- We will also move some of the content of the section *Turbulent fluxes at debris-covered sites and their controls* to the *Methods,* as indeed we described some of the methodology in that section.

We will streamline the text to emphasize the numbers that lead to our interpretations and conclusions. For example, instead of going through each energy flux individually in a systematic but dense manner in the section *Impact of the monsoon*, we now discuss the monsoon impacts in a more integrated way: We will start from the change in melt between pre-monsoon and monsoon (old Figure 6, old Table 5), then present the changes in the radiative budget before addressing the role of the turbulent fluxes and their changes. We will link each statement and number to the respective figure and/or table.

To further improve the readability of the *Results*, we will adopt a more intuitive language and terminology. For example, instead of using "sources" and "sinks", we will use "contributing to melt" or "reducing melt", in order to reduce confusion around the direction of the fluxes and their changes.

We hope that this new structure and writing style will allow us to explicitly draw together the distinct numerical results to answer our research questions in an easily understandable way. As an excerpt from the revised results section *4.4. Impact of the monsoon* (Note that the Figure numbers do not correspond to the original submitted version and that *M* will replace *dQ* as the energy available for melt):

**4.4.3 Impact of the monsoon on a thinly debris-covered glacier**

In contrast to the glaciers with thick debris, during the monsoon, the melt energy $M$ increases considerably at the thinly debris covered Hailuogou glacier. Although $SW_{net}$ contributes less energy for melt during monsoon and $LW_{net}$ remains overall small at this site (Figure 5), $M$ increased by $28.7\ W\,m^{-2}$ on average (Table A2), and mostly during the nights (Figure A10). The increase in melt energy is mostly driven by the turbulent energy fluxes: $H$ increases by $15.6\ W\,m^{-2}$ and $LE$ increases by $24.5\ W\,m^{-2}$ (Figure 5 and Table A2), with higher increases during the nighttime than during the daytime (Figure A10). While they acted to reduce melt at the glaciers with thick debris cover, here the turbulent fluxes drive additional melt during the monsoon.

Discussion:

We will restructure the *Discussion* in subsections that answer the new research questions and link the revised sections, figures and tables in the *Results* to the *Discussion* clearly.

As the *Limitations* and *Future work* sections might have distracted from the main outcomes, we will remove both sections and move some key elements of *Limitations (*i.e. on the debris parameters and moisture interception), to the *Methods*. We will also remove the *Implications* section but keep the most important messages on the possible climate change impacts for the *Conclusion*. A snippet from the *Discussion* section *5.3. How does the monsoon change the glacier surface energy balance*:

**5.3.3 Glacier with thin debris**

At the site with thin debris, we observe a melt-enhancing effect: the dark debris surface absorbs almost 90% of $SW_\downarrow$ in the case of Hailuogou (Table A2). With a short conduction length ($1\,cm$), the energy influx goes almost entirely to melt. Melt additionally increases during monsoon: higher wind speeds enhance turbulence resulting in an increase in $H$ (Section 4.5 and Table A3). Warmer and more humid air increases $LE$ inputs from condensation at the cold surface (Table A3 and Figure A8). Both turbulent fluxes thus become potent sources of melt energy (Section 4.4.3). This adds detail to prior observations and modelling results, that thin debris causes higher melt rates than at both clean-ice sites and sites with thicker debris cover (Östrem, 1959; Reznichenko et al., 2010; Reid and Brock, 2010; Fyffe et al., 2020), especially in humid environments (Evatt et al., 2015), e.g. the location of Hailuogou glacier.

Conclusions
We will rewrite the *Conclusion* so that it provides structured answers to the research questions.

Most of the figures themselves are overwhelmingly complex and the main points are not supported by them. Perhaps the manuscript can be more logically structured and extensive work can be done to give the reader a thread to follow.

To address the referee comment, we have restructured the manuscript in the way described above and introduced important changes to the main figures or removed some of them:

1) After careful consideration of the referee's comments we will modify the original Figure 6: we will add to the original figure panels (h)-(j) below, which depict the pre-monsoonal and monsoonal fluxes, their actual direction, their magnitude and their changes from one season to the other with the actual values from one site. The new panels will support the interpretation of panels (a)-(g), and should avoid confusion around the direction and magnitude of flux changes. We have made a panel for each surface type, and the numbers used are from one site for each of those surface types.

[Figure]

***New results figure. (a)-(g)*** *Differences in energy balance components from pre-monsoon to monsoon at each site including their uncertainties (error bars). The direction of change is to be considered relative to the sign of the original flux (x-axis). For example, a positive change in a negative flux means a reduction in the flux, and can also lead to a change in sign. Background indicates the surface type of the site: grey indicates debris-covered, light blue indicates clean-ice sites, and grey-blue indicates thin-debris.; **(h)-(j)** Alternative depiction of the changes from (a)-(f), summarizing surface types; Example Δ-flux numbers in [W m⁻²] refer to (g) Parlung No.4, (h) Lirung and (i) Hailuogou; Numbers for the other glaciers can be looked up in Table 5.*

2) To link the discussion on the *Impacts of the monsoon* to the respective results, we will introduce a new figure in the *Discussion.* The idea of this figure is to summarize the flux changes between the different surface types in a visually more straightforward manner.

[Figure]

***New discussion figure.*** *Triangles pointing down/up indicate a positive/negative flux with regards to our sign-convention, where positive/negative means a flux towards/away from the surface. Red/blue indicate an increasing/decreasing value of the flux when moving from pre-monsoon to monsoon. When signs switch, the underlying, empty triangles indicate the pre-monsoonal direction of the flux, while the overlying, colored ones indicate the monsoonal flux.*

3) We will move Figure 9 with the corresponding text (originally in section 5.2 *Sensitivity of seasonal flux changes to elevation and debris thickness)* to the supplementary material. These numerical experiments were intended to demonstrate that the seasonal flux changes are robust and do not depend on the actual elevation or debris thickness of the AWSs, but may have interrupted the flow of the  results and discussion.

To this point I found that the most compelling explanation of the role of differences in local climate came from the ERA-5 output and figure 3. But I must ask: What do the in situ station data tell us that the ERA-5 output do not already inform us about? There is quite a lot of scatter between the in situ site data (the data is from different years, elevations, surfaces, and aspects) unlike the patterns shown in the ERA-5 output.

We thank the reviewer for their perspective. Reanalysis data are extremely useful for many purposes, including catchment-scale hydrological modelling or even for the forcing of glacier-scale energy balance models of large glaciers. Here, we examined the ERA5-Land outputs to put our AWS records (which span only individual years) into their long-term context, as explained later in this answer. In fact, as shown for our study site Langtang, the reanalysis data captures the seasonal cycle of most variables reasonably well (Figure below). However, if we are interested in the monsoon impacts on the glacier surface energy balance and in the detailed processes behind them, we do not think that the accuracy of the reanalysis data is sufficient to reach our research objectives. The figure below makes evident that there are considerable local biases in each meteorological variable at our Langtang glacier site. Indeed, a few °C of air temperature bias (here, 4°C) or different wind speeds (>100% bias) particularly affect, and can even change the direction of the turbulent fluxes, which are key fluxes in the seasonal transition. These biases exist because first, a 9km grid element over high mountain terrain can integrate an altitudinal range of several thousand meters, as well as glaciers, snow cover, vegetation, surface water, and bare rock. Second, there are glacier-atmosphere interactions that create non-average conditions over the glacier surface, e.g. a colder boundary layer and katabatic winds. Those processes cannot be represented in sufficient detail by current climate models and reanalysis products. Third, climate models are known to not perform well in regions with complex topography and where local observations are scarce.

The AWS data, on the other hand, allow us to reproduce the glacier surface energy balance accurately, make inferences about the surface (debris) properties, and evaluate the model performance. The referee makes a valid point that our study site records have different duration and refer to different years, which might complicate their comparison. However, very few on-glacier datasets are available in High Mountain Asia because they are very difficult to collect, and therefore they rarely overlap spatially and temporally. Importantly, the major result of our analyses is that, despite the differences between sites, there are common patterns in the seasonal changes in energy fluxes. To make sure that we do not accidentally compare exceptional years, and draw the wrong conclusions from that, we indeed put our records into the context of average conditions by comparing them to the ERA5-Land data. This showed that the seasonal variability is greater than the interannual variability for all variables and across study sites, and that the years of our records represent typical conditions.

In response to the reviewer's comment, and to avoid ambiguity as to what we use the ERA5-Land data for, we will move most of the description of the reanalysis data, including the corresponding Figure 3, to the supplementary information, and make clear in the main text that we use those data and figure only to show that our selected years are representative of the multi-annual patterns.

[Figure]

*Figure. Monthly sums (precipitation) and mean (all other variables) of ERA5-Land vs. Langtang on-glacier weather station data;*

Perhaps the figures and text can more clearly show the take homes from the station data and support the more general take homes?

To link the key outcomes better to our analysis and figures, we will restructure and modify our manuscript and some of the figures in the ways described above.

My sense is that this could be an interesting, valuable study for TC but as it stands I am not sure if the analyses actually support the conclusions and if using in situ station data is better suited for this question than atmospheric reanalysis output.

We thank the referee again for appreciating the potential value of our study. We tried to respond to the referee's concerns in the best way possible, and will revise the manuscript considerably based on the comments. We will link our conclusions to our results more explicitly in the text, and have made it clear in an answer above why it was necessary to use station data rather than reanalysis data for our study.

More specific comments:
Line 30-32 dates on Mölg should be 2012,2014 and the references should be in order of date in line 33 with the oldest first. Should be corrected throughout.

Thank you for these suggestions, we will revisit this citation and sort citations throughout the manuscript by date.

Line 99. too may uses of 'extensive' in this paragraph.

We will fix this issue and revise a part of this paragraph in order to streamline it.

Figure 1. I cannot see the RGI glaciers in panel A. Please change the color of the glaciers. The arrows in panel A seem a bit inaccurate considering that the Indian summer monsoon certainty affects easter Nepal and too the west as well.

We will revise this figure based on these suggestions. We will give the RGI glaciers a more visible blue shade. We will change the arrows to represent the influence of the Indian Summer Monsoon more accurately. We will also rearrange panels b and c slightly and add a few elements that are missing: North arrow and coordinates with tick marks for panels b and c.

[Figure]

*Figure 1 (a) revised glacier color, monsoon influence, guides for glaciers; (b) and (c) scale bars and north arrows;*

Tables 5 and 6. Perhaps these should be in the supplement? They are rather overhelming to try to pull anything away from them.

We will move these two tables to the supplementary information

Section 5.1.1 Here many of these points are expected and reproduced by other studies. It seems to me those other studies should be cited here.

This is a good suggestion and we will add additional references. For example:
The importance of the radiative fluxes and their changes through monsoon were discussed at individual sites in a number of studies (e.g. Kayashta et al., 1999, Aizen et al. 2002, Yang et

al., 2011, Mölg et al., 2012). In studies comparing different sites, Zhu et al. (2018) and Bonekamp et al. (2019) identify the timing and quantity of snowfalls as major controls on the glacier mass balance through the albedo effect. Mölg et al. (2012) discuss in particular the role of spring snow accumulation and the importance of monsoon onset timing in controlling the seasonal mass losses. Fujita et al. (2000) highlight the important role of monsoonal summer accumulation, which we called 'ephemeral snow cover from monsoonal precipitation', in protecting the glacier through the albedo effect.

Aizen, V. B., Aizen, E. M., & Nikitin, S. A. (2002). Glacier regime on the northern slope of the Himalaya (Xixibangma glaciers). *Quaternary International*, *97*, 27-39.

Bonekamp, P. N., de Kok, R. J., Collier, E., & Immerzeel, W. W. (2019). Contrasting meteorological drivers of the glacier mass balance between the Karakoram and central Himalaya. *Frontiers in Earth Science*, *7*, 107.

Kayastha, R. B., Ohata, T., & Ageta, Y. (1999). Application of a mass-balance model to a Himalayan glacier. *Journal of Glaciology*, *45*(151), 559-567.

Mölg, T., Maussion, F., Yang, W., & Scherer, D. (2012). The footprint of Asian monsoon dynamics in the mass and energy balance of a Tibetan glacier. *The Cryosphere*, *6*(6), 1445-1461.

Yang, W., Guo, X., Yao, T., Yang, K., Zhao, L., Li, S., & Zhu, M. (2011). Summertime surface energy budget and ablation modeling in the ablation zone of a maritime Tibetan glacier. *Journal of Geophysical Research: Atmospheres*, *116*(D14).

Zhu, M., Yao, T., Yang, W., Xu, B., Wu, G., & Wang, X. (2018). Differences in mass balance behavior for three glaciers from different climatic regions on the Tibetan Plateau. *Climate Dynamics*, *50*(9), 3457-3484.

---

## Author Response (AR1)

**Answers to Referee #1**

We thank the referee for their useful comments. We answer all comments point-by-point below each statement in blue font.

Review on the manuscript entitle "Understanding monsoon controls on the energy and mass balance of Himalayan glaciers'

General comments:

Overall manuscript has provided a comprehensive study of the glacier energy and mass balance for seven sites and further generalized for the whole Himalaya. That may be the reason; the title has come up with Himalayan glaciers. However, this study has focused on the only on the seven glaciers and circled around the Nepal Himalaya and Tibetan Himalaya. So Eastern Himalaya is more appropriate.

We thank the referee for acknowledging the comprehensive nature of our study. The reviewer has a good point with regard to terminology. Our intention was not to generalize our results for the whole Himalaya and we are sorry if the title suggests this. We agree with the reviewer and changed the title to "Understanding monsoon controls on the energy and mass balance of glaciers in the Central and Eastern Himalaya". We think however that it is appropriate to use both Central and Eastern Himalayas, following Bolch et.al (2012), where "Central Himalaya"encompasses mostly the Nepalese Himalayas, and where "Eastern Himalaya" and "Western Himalaya" refer to the regions to the east and west of the Nepalese Himalayas, respectively. By moving "glaciers" in front of "Central and Eastern Himalaya", we hope to reduce the impression of generalization additionally.

There is several new information which is really valuable for the understating the summer accumulation type glaciers. One of that is: At all sites, ice melt is the dominant mass loss component, accounting for 65.8% (Changri Nup) to 95.4% (Hailuogou,) of the total mass losses.

We thank the reviewer also for appreciating the novelty of our results. Regarding the numbers that the reviewer cites and from which it is evident that ice melt is the dominant mass loss component: this is so as we have only considered the mass losses during the ablation period for which measurements are available (May to October for Changri Nup and mid-May to October for Hailuogou). The numbers for the year-round mass losses would include a greater snowmelt component. We modified the Results section of the manuscript in the way shown below, to make sure this is clearer to the reader (L323-330):

**4.1 Modelled mass balance**

The ablation season average melt rates vary considerably across sites: the highest value of  $42.7 \, mm d^{-1}$  is reached at the low-lying site with thin debris cover, Hailuogou, and the lowest value of  $6 \, mm d^{-1}$  is evident at Langtang, a site at moderate

- 325 elevation but with the thickest debris cover out of all study sites (Figure 4). The largest average seasonal mass loss component at all sites is ice melt, with a minimum of 65.8% of the mass losses at Changri Nup (Figure 4c) and up to 95.4% at Hailuogou, (Figure 4g). This is followed by snowmelt, accounting for only 0.1% at 24K (Figure 4e) but as much as 33.1% at Yala (Figure 4c) of the seasonal mass losses. Sublimation from ice and snow represents a very small share of the seasonal mass losses, and ranges from 0.01% (Lirung, Figure 4a) to 1.2% (Changri Nup, Figure 4d). It mostly occurs under dry conditions during
- 330 pre-monsoon at the highest sites (Changri Nup, Yala).

**Few more general comments, in fact it is query to be generalize.**

**The manuscript only talks about pre-monsoon and monsoon period, what about post-monsoon? Does it differ from pre-monsoon?**

We have also analysed the post-monsoon period and compared it to the other two seasons. The main reason why we did not include this analysis in the manuscript is that the AWS data for the post-monsoon were unreliable at the two highest sites (Yala after mid-September and Changri Nup after August), especially with respect to the precipitation and snow depth measurements. We were concerned that this would bias the resulting energy and mass balances. Although we had multi-year timeseries at hand, the chosen years for those two sites contained the most complete records of an ablation season. We have described this for Changri Nup in section *4.1 Modelled mass balance* and will also add this description for Yala. We also felt that the paper already contains extensive results that allow identifying the distinct characteristics of the monsoon. Since Referee 2 noted that the paper is already dense and contains much information, and both figures and text would become overly complex if we also added the post-monsoon (this necessitates two additional comparisons, pre/post and monsoon/post), we decided to refrain from including an analysis on the post-monsoon into the main text, and hope that the reviewer will agree with us.

(ii) There is no discussion about the effect of winter precipitation on the energy and mass balance of the glaciers. Although the manuscript deals with understanding monsoon controls on energy balance and mass balance, but winter precipitation has equal control over the energy and mass balance.

We fully agree with the concerns the referee raises: the timing and quantity of winter and spring snowfalls greatly shapes the **annual** energy and mass balance through the albedo effect. However, while an analysis of the influence of winter precipitation would be a worthwhile analysis, it goes beyond the scope of the present study, which focuses on identifying the influence of monsoonal conditions on the ablation season energy balance.

**Section wise comments:**

L2: "large temperature amplitudes" make it simpler like large temperature ranges.

This is a good suggestion and we modified the text.

L5-6: This sentence, I would like to see at the end of the introduction, where citation of work may validate it.

We agree that this might not be an appropriate sentence for an abstract. As the last paragraph of the introduction started with a sentence of similar content, we removed it from the abstract, and instead stressed the importance of energy balance studies in a shorter sentence (L3-4). *"Glacier energy and mass balance modelling using in-situ measurements can offer insights into the ways in which surface processes are shaped by climatic regimes".*

**L7: 'Himalayas' it is for curiosity on using 'The Himalayas' instead 'The Himalaya'. I am actually not sure which one is better.**

We shared the reviewer perplexity here. Looking this up, according to the word origin (Sanskrit, "hima" = snow, "alaya" = abode), the name refers to the mountain range as an "abode of snow". Thus, from the etymological perspective, the singular "Himalaya" is more appropriate. In modern times, it was misinterpreted as referring to the single mountain, hence all the Himalayan mountains together were turned into the plural "Himalayas". In published cryosphere literature, both writings are frequently used, so we decided to use the etymologically more correct way, and switched to "Himalaya" throughout the manuscript. We thank the reviewer for this hint!

L 19: "dirty-ice glaciers", somewhere it was mentioned as thin debris, so does the dirty-ice glaciers are the same ?. If so then thin debris is mostly lies over the patches or around the higher elevation. Whereas, it has mentioned here as dirty-ice glaciers, which what I understand is that the whole glacier has dirty-ice only.

We revisited these definitions and realized that "dirty ice" may after all not be the right term to describe the surface of Hailuogou glacier's ablation zone. According to Fyffe et al. (2020) dirty ice is only "partially debris-covered", "patchy" or "discontinuous". According to our own field-observations, Hailuogou's ablation zone is to a large extent continuously covered with a thin layer of fine clasts and scattered with coarser clasts, which would leave the thin layer visible, and directly influenced by the atmosphere. Co-author Liu Qiao, who has maintained an AWS on Hailuogou between 2008 and 2013, has measured a debris thickness of 1cm at the AWS site. We have therefore decided that using the definition "thin debris" is more appropriate and removed the use of "dirty ice" everywhere. We also revised the description of Hailuogou glacier in L105-113 and removed the mention of dirty ice in the *Introduction* section and a related citation in L55-59.

**L 21: (Yang et al., 2017), please check.**

We removed this. This is an artefact in our LaTeX code.

L28: "Karakoram, Pamir and Kunlun ranges in the east". I think it should be 'west'.

We corrected this mistake.

L55-57: This has no information except to show that these researchers have published work on debris-covered glaciers.

We respectfully disagree here, because this citation lists all the studies introducing energy balance models for debris-covered glaciers and thus represents the evolution and state of the art of this type of model.

L62-63: In continuous to the pervious comments. Here are some other references having in situ observations on the central Himalayan glaciers with the perspective of debris cover and thickness influences on ice melt (Shah et al., 2019 and Pratap et al., 2015).

Thank you for these suggestions. We already cited Shah et al. (2019), who conclude that debris thickness has a stronger control on glacier melt than elevation. We however missed Pratap et al. (2015) and now also include this study in L54-55.

L73-75: this whole paragraph, I dint see any sense before to define the objectives of this study.

We agree that this part seems disconnected and interrupts the flow of the introduction. It does also not contain essential information for motivating the analysis, so we decided to remove it from the text.

L87: 'glacierised' I generally practice to use 'glacierized' as per Cogley et al., 2011 (glossary of glacier mass balance and related terms).

Both the US American ("glacierized") and British ("glacierised") spellings are accepted and used in the literature. We decided to generally adopt English spelling in the manuscript and thus kept "glacierised".

L92: Table 2 cited before Table 1, check it with journal style.

Thank you for spotting this. We changed the order of these two tables.

L104: This might be the ablation area that has disconnected from the accumulation area. if this is the case then in the Table , the Lirung Glacier's characteristics needs to be revised.

We are not sure we understand the reviewer's comment here and would kindly ask him/her to clarify how we should revise Lirung glacier's characteristics in Table 2. Currently, the table contains the characteristics of both the accumulation area and the (dynamically disconnected) ablation area together, e.g. the sum of the areas of both glacier parts. We now made this clear in the caption:

Table 2. Characteristics of the study sites. Planimetric glacier and debris surface areas, mean elevation, slope and aspect were calculated using the updated Randolph Glacier Inventory 6.0 by Herreid and Pellicciotti (2020) and the USGS GTOPO30 digital elevation model. Slope and aspect are mean values for the whole glacier. MI ('Monsoon-Index') is the mean June-September portion of the ERA5-Land total annual precipitation (1981-2019); For Lirung, where the ablation zone has dynamically disconnected from the accumulation zone, the glacier characteristics represent both zones together.

Figure 2: Caption: "(blue bars)" For me the color is aqua and not blue. "area on the x-axis [km2] and altitude on the y-axis [m.asl]", This information isn't shown in the figure. Area (size) of the glaciers is not clear, therefore additions of a scale bar and direction arrow is required. "Black crosses" this sign need to change as at Yala Glacier it entirely covers the glacier. Make it red dot with AWS on the side as a legend.

We agree that "aqua" is a more suitable name for the colour. We also changed the colour name for the area/bars indicating debris cover to "olive". The glacier area is expressed in 100m elevation bands in the diagram. We agree that it is not easy to judge the glacier size and orientation without a scale bar and direction arrow. We now added these elements to the figure. We also decreased the size of the x-indicators and arrows for better readability and added a legend, but we kept the black color for contrast and style reasons.

Figure 2 revised

**L134: The figure description is not in order.**

We will change the sequence of the text, so that the references (a-e) are called in order. Based on comments from Referee #2, we moved this part "*Climatic and meteorological conditions*", including the Figure itself, to the supplementary.

L138: 1st if one consider the lirung and yala glaciers with an elevation difference 1000 m asl in the same catchment, and 2nd by including the fully debris covered ablation area and other clean ice , how it can be justify that the mean monthly 2 m Ta is very similar on the both sites. Though, it is an observation (Fig. 3a) but just to rethink.

We thank the referee for this useful and insightful comment. The referee is very right that the similarity in air temperatures between Lirung and Yala glaciers is unrealistic for precisely the reasons the referee mentions. But please consider that the climatology for each glacier in Figure 3 is taken from one gridcell of 9x9km horizontal resolution of the ERA5-Land reanalysis product and represents average conditions within this gridcell, and not the conditions at the actual elevation of each glacier. We refrained from adjusting the ERA5-Land outputs, as the purpose of this figure is only to show that the study year for each site falls well within the typical interannual range – these data would need careful downscaling to adequately represent on-glacier conditions. We had already acknowledged this circumstance in L130-133 (old version): *"Here, we use the monthly averaged ERA5-Land reanalysis data (Muñoz Sabater, 2019) to provide an overview of the long term climatic patterns, … and evaluate the representativeness of the AWS records in terms of*

seasonal variability ..., while acknowledging that the absolute values from the reanalysis dataset might be biased."

Motivated by the referee's comment, we reformulated this sentence and added to it a sentence on the representativeness (L115-119). In the new version, we also moved most of the detailed description to the Appendix, based on a comment by referee #2:

We use the monthly averaged ERA5-Land reanalysis data (Muñoz Sabater, 2019) to evaluate the representativeness of the AWS records in terms of seasonal variability (Figures A3 to A9), and to provide an overview of the long term climatic patterns, e.g. the average monsoonal regime from June through September (Figure A1). We thereby focus on the qualitative aspects, given that the absolute values from the reanalysis dataset are not representative for the AWS location at the glacier surfaces. A detailed description is given in the Supplementary.

**L192: "surface temperature Ts". Please elaborate that how Ts was calculated?.**

The reviewer is right that our formulation was confusing. The calculation of surface temperatures was explained in L176-179 (old version) "To close the energy balance, a prognostic temperature for the different surface types  $(T_{sno}, T_{deb}, T_{ice})$  is estimated for each computational element. Iterative numerical methods are used to solve the non-linear energy budget equation until convergence for the ice and snow surface, and the heat diffusion equation for the debris surface, while concurrently computing the mass fluxes resulting from snow and ice melt and sublimation."

We did not inform the reader however that  $T_{sno}$ ,  $T_{deb}$ ,  $T_{ice}$  are equivalent to  $T_s$  in the equations that are not specific to a surface type.

We now added a sentence clarifying the use of the symbols (new version L148-150) :

"In the case of snow, debris and ice surfaces,  $T_{sno}$ ,  $T_{deb}$  or  $T_{ice}$  are equivalent to the element's overall surface temperature  $T_s$ . In the following, we use the surface type specific symbol for surface specific equations, while we use  $T_s$  for equations valid for all three surface types."

**3.2 Mass balance in T&C. if it is the same name used before, i would suggest to use T&C model throughout.**

We changed to use "the T&C model" everywhere.

**L312: delete 'We vary'**

Unfortunately, we do not understand why "We vary" would be unnecessary here. As there might be a confusion around the word "vary", in the revised version we use the word "perturb" instead (L295).

**L340: choose other word as it was already used with Tibetan plateau.**

We tried to find an alternative, but found no other word that would describe our observation as precisely. "plateau" is used as a verb here, while in "Tibetan Plateau" it is used as a noun or name.

Figure 4. Caption and legend.

Measured and Obs., change to single. Black circles seems to be black dot.

Thank you for pointing us at these inconsistencies. We made the changes in the caption and legend.

Figure 5. (i) what is the reason for using different color scheme for same component. I cannot differentiate the ice melt and sublimation for the LIR glaciers. I think use of single color like for LAN glacier would be ok.

This is a good question. We gave each study site its own color signature throughout the manuscript. We were hoping that this would help the reader to intuitively recognize the study site by color in addition to the name. We had the experience in earlier studies, that using only the name of several study sites would sometimes confuse the reader, and the reader would have to spend extra time to repeatedly relate the name to e.g. the geographic location.

We changed the color indicating sublimation to allow for an easier differentiation between sublimation and ice melt.

---

## Referee Report (RR1)

Review of manuscript "Understanding monsoon controls on the energy and mass balance of glaciers in the Central and Eastern Himalaya" by Fugger et al.

**General comments**

This manuscript presents measurements and modelling of surface energy and mass balance at 7 sites throughout the central and eastern Himalaya with a view to assessing the how debris cover modifies the effect of the monsoon on glacier fluxes. On-glacier automatic weather station records from the ablation zone of each glacier and a land surface model are used to derive energy and mass fluxes over 4-6 months. The land surface model includes both snow and ice volumes as well as debris cover of various thickness on 5 glaciers where debris cover is observed. The land surface model is optimised and validated against observed melt and surface temperature. The seasonal variation of energy fluxes is presented and differences between pre monsoon and monsoon periods assessed. The commonalities and differences between sites are discussed, especially the role of debris cover in creating differences. The synthesis of 7 sites from a wide geographic area, and analysis in a common framework are a particular strength of the manuscript.

The paper is generally well written and has follows a logical progression. The methods are appropriate and are mostly well described. The results provide new insights into the processes controlling mass loss across these sites. Most statements in the discussion/conclusion/abstract are supported by the results.

The main conclusion that is not shown by the current results is the assertions about lower climate sensitivity for debris covered compared to clean ice glaciers (line 14, 531-542, 572). These results are not demonstrated and should be presented as speculations or hypotheses. A formal analysis of climate sensitivity could be made with this dataset and model, and perhaps this is an area for future research.

The authors appear to have addressed most of the reviewer comments, however further discussion of the role of post monsoon and winter precipitation (Reviewer #1, general comments) in controlling mass balance should be included in the discussion, especially if the authors wish to discuss the significance of the present results (monsoonal controls) on future mass balance i.e. to what extent do monsoonal, pre/post monsoonal and winter mass fluxes determine annual mass balance?

While the authors have reordered the manuscript in response to Reviewer #2, too much material has been placed in the appendix and the appendix needs some more organisation. Some figures and tables could come into main body of text (e.g. diurnal patterns), and the section (text, table, figure) on sensitivity to elevation / debris cover thickness. Organising the appendix into sections with related figures/tables together and text to give context would greatly help the reader.

Further specific comments are given below, but some main points to address are highlighted here.

- Some discussion of the limitations of the study is warranted – this should include the fact that all records cover only one season (how significant is interannual variability?) and that the sites are in ablation areas and that surface energy and mass fluxes (along with their response to the monsoon) will change at higher elevations.
- The sign convention and terms used for changes in fluxes needs clarifying at times
- The methods and results used to regress the turbulent heat fluxes against meteorological forcing needs more description – it looks nice but is hard to interpret meaning of these results.

- The importance of turbulent fluxes at clean ice sites is downplayed more than needed - the change in latent heat flux is similar at clean ice and debris cover sites.

With some reorganisation, clarification to methods and corrections to text this manuscript will make a valuable contribution to the literature.

**Specific comments:**

131 – the sign convention in Equations 1-3 does not follow that stated in the text - i.e. "The sign convention is such that fluxes are positive when directed towards the surface (line 143)." Thus, all terms should have a + sign in front of them. The exception is M, which is treated as a positive term throughout the text, so should retain a minus sign or appear on the right hand side of equations 1 and 3.

Table 2 – columns describing the mean air temperature, wind speed, RH, precipitation etc of the sites would be very useful in understanding differences in SEB components between sites, particularly the turbulent fluxes.

181 – a brief description of $r_{ah}$, particularly how it relates to deltaT, Ws and $z_{0m}$, $z_{0h}$ is warranted here given the key role the turbulent fluxes play in the analysis and conclusions.

184 – the assumption of $z_{0h} = z_{0v} = 0.1 z_{0m}$ needs supporting, particularly for large $z_{0m}$ over debris cover where this ratio may become smaller.

211 – the discretisation of subsurface layers in snow/debris/ice is ambiguous - please provide a clear description of the number of subsurface layers used to solve the conduction equations

252 – please introduce the term *In* here (used later but not defined) and provide further details on how the calculation of debris SEB is altered by this term.

290 – please provide NSE (of melt and Ts) for each site for both steps of the optimisation procedure.

Figure 3 – albedo and precipitation observations indicate the YALA glacier had consistent snow cover in the pre-monsoon period, but this is not shown in the modelled results – was this the case, and if so, how might this discrepancy affect the results for YALA?

323 – it is unclear how was the monsoonal period was identified in each record (Figure 3,4). Please add this detail and some discussion of how sensitive the results are to this choice.

323 – also, how were the individual years chosen from the multi-year records at each site? and how sensitive are the results to the chosen years?

355 – some comment on the direction of turbulent fluxes for clean ice glaciers would be useful here

361 – "Reflected shortwave radiation SW↓, which removes energy from the surface, and which is controlled by the surface albedo, follows these changes (Figure 6), becoming less negative by +5:4 (24K, pre: -18:5, mon: -13:8) and up to +164:8 (Parlung No.4, pre: -219:6, mon: -54:8) between sites." This statement is ambiguous – do you mean that the changes in outgoing shortwave follow changes in albedo, or in incoming shortwave? If you mean the later, then this statement is not strictly true – the changes in outgoing shortwave radiation are dominated by changes to albedo at Parlung No 4. Please revise.

364 – "an increase of the flux" – ambiguous (see comment for Figure 8). Suggest changing to" where SW↓ becomes more negative -12:1W m-2 (pre: -60:6, mon: -72:7), as a consequence of…"

374 – some comment on which term in SWnet is causing the increase melt would be useful here.

383 – "glacier-cooling H becomes a smaller flux" – ambiguous (see comment for Figure 8). Suggest "becomes less negative"

385 – "change in LE partly offsets the changes in H, with increases in the flux ranging from…" – ambiguous (see comment for Figure 8). Suggest "LE becoming more negative by…"

387 – following you sign convention (energy input to surface is positive), H is increased (i.e. less negative) and LE is decreased (more negative) in monsoon period. Also, as the changes in radiative and turbulent fluxes do not balance separately, it would better to state that that "reduced SWin and more negative LE are balanced by increased LWin and less negative H."

Figure 6 – nice figure, but why choose to present only one glacier in the alternate depiction, and not the average of all sites of that type?

397 - Figure A11 – this is an interesting and informative figure and should be described in the main body of the text (Section 4.4) for all surface cover types.

404 – the regression model needs a much better description in its own section of the methods. It is unclear what the timescale of regression is (hourly/daily/weekly/seasonal) and what equations are used to fit the model. Without this it is hard to interpret the meaning of these results. Please revise methods and results.

413 – "Neither RH, gT , or Ws on their own, nor their combination explain the variability of LE across sites with thick debris" – this analysis is not shown here. Please revise

Table A3 – this could be placed in the main body of the manuscript to support these results.

456 – "the turbulent fluxes play a minor role on the clean-ice glaciers" – this statement only applies to sensible heat, as changes in turbulent latent heat fluxes are similar magnitude on clean ice and as debris covered glaciers. Please revise.

489 – this is the first mention of the elevation and debris cover thickness sensitivity experiment and comes as a surprise. The sensitivity experiment could be worked in the main body of the manuscript (methods and results) or at least should be mentioned in the results section.

Figure 8 – the sign convention does not follow the same logic as previous figures: 'increasing value' is ambiguous, as it can mean a more or less positive flux. Better to use 'increasing magnitude' along with the sign of the flux, or be specific and use 'positive change' and 'negative change' to refer to changes that increase and decrease the energy available for melt. Please revise here and throughout section 5.3.

535 – "In contrast, the turbulent fluxes 'work for' the glaciers with debris above the critical thickness, and the melt-equalizing effect of debris under monsoon (Section 4.4.2) would likely remain in place". and 540 – "Here we confirm this hypothesis [that mass balance of debris-covered glaciers might be less sensitive to climate warming than clean-ice glaciers]" – This effect remains speculative as the sensitivity has not been tested. While the current study does highlight the different roles of turbulent heat fluxes over debris vs clean ice and how these change in monsoonal conditions, it doesn't assess how this will change in the future. There may be other interactions that change the role of different fluxes in the future e.g. if RH also increases in the future, then the magnitude of evaporative losses during the monsoon may decrease, thereby further increasing the

melt experienced under debris compared to present day. The dataset presents an opportunity to test the sensitivity, but this is not presented here. Please revise.

554 – 'modulated' -> 'increased'?

556 – the cold surface also favours sensible heat exchange into the glacier surface – please revise.

557 – "melt-rates increase compared to the pre-monsoon at the clean-ice glaciers" – this is not significant for Yala – please revise.

563 – "The cooling induced by H at the same time decreases, with the result of unchanged available melt-energy M during monsoon." Increased LWin during the monsoon also helps offset the decreased SWin along with H. Please revise.

570 – it is unclear how the results support a reduced climate sensitivity at debris covered sites. Please provide additional material in the results/discussion or revise.

Section A1 – this could be presented in the main body of the text (methods/results). Also, some features are ambiguous – i.e. was air temperature the only variable modified for elevation (i.e. not incoming longwave radiation and RH?

Figure A12 –Please modify caption as the fourth sentence infers debris cover was varied for all glaciers - Presumably the debris cover thickness was only varied for debris covered glaciers? Also the first sentence could be more instructive e.g. "Sensitivity of changes in the individual fluxes when moving from premonsoon to monsoon, to elevation and debris cover thickness (for debris covered glaciers only). "

Table A3 – the derivation of cloud cover fraction should be described in the methods section.

**Editorial comments:**

347 – "snow free" -> "snow-free"

350 – "re-emitted" it is perhaps better to say 'lost' as turbulent fluxes do not 'emit' energy, so to speak.

385 – missing negative sign from "to -24.4"

Figure 7 caption. As HAI is excluded from panel (b), the commas should be removed from the last sentence, so it reads "Only debris-covered glaciers where LE is a glacier-cooling flux are shown"

Figure A10 – appears to be missing or perhaps figures need relabelling.

---

## Author Response (AR2)

**Author's response Revision 3**

**Understanding monsoon controls on the energy and mass balance of glaciers in the Central and Eastern Himalaya**

Stefan Fugger, Catriona L. Fyffe, Simone Fatichi, Evan Miles, Michael McCarthy, Thomas E. Shaw, Baohong Ding, Wei Yang, Patrick Wagnon, Walter Immerzeel, Qiao Liu, and Francesca Pellicciotti

**Response to the editor and general revision**

**Dear Editor,**

We are pleased to resubmit our revised manuscript [tc-2021-97] "Understanding monsoon controls on the energy and mass balance of glaciers in the Central and Eastern Himalaya", which we revised after receiving this second round of reviews.

Reviewer 3 had no major comments, but a number of specific comments, of which some overlapped with Reviewer 4's main comments, which were: (i) That the implications of our results under future climate were overstated while not supported by the analysis, and that those implications should be presented as speculations or hypotheses. (ii) That the role of post-monsoon and winter mass-fluxes should be included in the discussion and more generally (iii) that a discussion on the limitations of our study is required, in particular: the single-season perspective of our analysis, and that all results were derived for the glacier ablation areas only. (iv) That the Appendix needed restructuring and that some content could be re-introduced into the main text of the manuscript. (v) That some elements, such as the regression models used or the calculation of aerodynamic resistances, needed a better description.

We would like to thank Reviewers 3 and 4 for their thorough and constructive comments, which further contributed to improving the manuscript, and which we have addressed below. We summarise here the most important modifications, and then provide a point-by-point answer to the two reviewers comments.

**In summary, we have**

- revised the text parts around the implications of our results for Himalayan glaciers in a changing climate, in order to not overstate them, and in order to avoid creating the impression that those were our main conclusions.
- added a section on the limitations of our study, where we discuss our focus on individual ablation seasons and explain why we could not expand our perspective to the post-monsoon and winter seasons. In addition to that, we identified key knowledge gaps and gave recommendations for future research.
- replaced the Appendix with a better structured Supplementary material and moved some of the text and figures from the Supplementary into the main text
- revised and chosen carefully our wording when presenting changes in energy fluxes in order to avoid confusion around the direction of fluxes and their changes.
- revisited the regression analysis performed in order to understand the controls on the turbulent fluxes and added to the Supplementary Material more detailed information and figures around this analysis.

- Highlighted the importance of turbulent fluxes on clean-ice glaciers, including their changes in the seasonal transition, while they were mostly discussed for the debris-covered glaciers in the previous version of the manuscript.
- added to the Supplementary Material more detailed information on aerodynamic resistances and aerodynamic roughnesses.
- added to the Supplementary Material a more detailed description of the definition of monsoon onset and cessation.
- recalculated all uncertainties with wider uncertainty ranges on the radiometer measurements.
- added all missing numbers and variable definitions requested.
- implemented minor reformulations throughout the text and adjustments to numbers and figures following the two reviewers comments.

As a result of these changes, we have added to the Results a subsection on the *Sensitivity of seasonal flux changes to elevation and debris thickness* and to the Discussion a new subsection on *Limitations*. In order to reduce the manuscript in volume, we have replaced the former Appendix with a new Supplementary Material, provided in a separate document, which we have structured into 6 thematic sections. We have moved two figures from the Supplementary Material (former Appendix) to the main manuscript and added two new figures and one new table to the Supplementary Material.

We very much hope that the revised manuscript is now appropriate for publication in The Cryosphere.

**In response to Reviewer 3**

**General comments**

The authors present a study of surface energy and mass balances during different climatic regimes, the pre-monsoon and the core monsoon season, at different glaciers in the Himalaya. Interestingly, the glaciers represent sites with thick, thin or without debris cover. From observations and modelling at the point scale the authors derive the energy and associated mass fluxes and show, that depending on the surface type, the monsoon influence is either minimal or enhancing melt compared to the pre-monsoon season.

The manuscript is long and dense but the structure is well built and makes it easy to read. The hypotheses are clearly laid out and the methods are well explained. The discussion section brings up the questions asked in the hypotheses and gives concrete answers with a generalized figure (Fig 8) as overview of the regime changes and impacts on different glacier sites. Figures are coherent and help the reader understanding the key points. The substantial supplementary material supports data and results.

Overall, I rate the manuscript ready for publication after clarifying some minor comments.

We would like to thank the reviewer for their detailed review and positive evaluation of our manuscript. We provided reference to line numbers of the revised manuscript, unless otherwise stated. In order to make easier links between related comments within this document, we numbered the comments in the form R3.x (comments of reviewer 3) and R4.x (comments of reviewer 4).

**Specific comments referring to manuscript version 3:**

R3.1: L 131, Equations 1-3: I'd suggest to generally write the energy balances as addition, i.e. without any minus symbol. All fluxes (except SWnet) can switch sign and the sign they receive in the equation depends on the sign convention used (and explained in L 142). According to this, M must receive a negative sign as it denotes the available energy flux for melt, which removes energy from the surface by phase change. This has implications on the sign of M in several figures and tables.

We agree with this, and have changed the three equations accordingly! (Eq. 1-3) We also adjusted the signs of M to negative in all equations, figures and tables.

R3.2: L 140: I suggest a rephrasing. In the given form of Eq 1, M is the available energy for melt, not the net energy input into the snow or ice. We just make it the net "input" by solving the equation for M, but note, the sign must be negative according to convention in L 142.

We agree and rephrased this part according to your suggestion, since M is the net input minus the energy needed to heat the snow and ice pack. (*L* 136-137) For the sign of M, please see above (R3.1).

R3.3: L 177, section 3.1.4: All equations and parameters are well defined. I just miss an explicit explanation of the aerodynamic resistances (rah and raw) and their control of atmospheric stability. This could go into the supplementary material.

In response to both reviewers on this issue (see also R4.12), we have added a Section S3 Aerodynamic resistance and aerodynamic roughness to the Supplementary Material, where we describe our implementation of a simplified solution of the Monin-Obukhov similarity theory. The derivations of rah and raw are expressed in the formulae. In addition we have added references to the new supplementary section in Methods section 3.1.4 Turbulent energy fluxes as follows: (L 178)

"The aerodynamic resistance r\_ah [s m^-1], (Note: which in our model is equal to r\_aw) is calculated

using the simplified solution of the Monin-Obukhov similarity theory proposed by (Mascart et al., 1995) and implemented in (Noilhan and Mahfouf, 1996), for details see also supplementary Section S3."

R3.4: L 198: Controls on turbulent fluxes. I'd remove this paragraph here and put it into the supplementary material as well with a bit more information on the regression. For now, this information is a bit lost and becomes relevant in Fig 7a only, where a reference to the supplementary material could be made.

We thank the reviewer for this input. Both reviewers requested more information on the regressions (Reviewer 4's comment R4.29), so we introduced new supplementary figures, visualising the individual scatter plots and regression models, including *adjusted*  $R^2$  values and model formulas in the plots (see Figures R1-3 below). We removed the text from the Methods and added it to the Supplementary Material (Section *S5. Controls on turbulent fluxes)*. There, we also added the formulae for the three predictors, the general formulation of the regression models and the timescale used. We added a reference to this section to the caption of Figure 8a. We improved the regression by removing values of  $LE = 0 W m^2$ , which corresponded to timesteps when the debris water interception storage is empty during evaporation conditions and no latent heat flux is possible. Upon revisiting this part and based on a comment by Reviewer 4 (R4.30), we then also replaced the temperature gradient *delta\_T* with the vapour pressure deficit *vpd* as a predictor in the regression analysis for LE.

Figure R1: Regression of turbulent fluxes against temperature gradient *delta\_T* (H) and vapor pressure deficit *vpd* (LE) for the debris cover sites (Figure S10)